# A FN-MdV pathway and its role in cerebellar multimodular control of sensorimotor behavior

Xiaolu Wang[1], Si-yang Yu[1], Zhong Ren[1], Chris I. De Zeeuw [1,2✉] & Zhenyu Gao [1✉]

The cerebellum is crucial for various associative sensorimotor behaviors. Delay eyeblink conditioning (DEC) depends on the simplex lobule-interposed nucleus (IN) pathway, yet it is unclear how other cerebellar modules cooperate during this task. Here, we demonstrate the contribution of the vermis-fastigial nucleus (FN) pathway in controlling DEC. We found that task-related modulations in vermal Purkinje cells and FN neurons predict conditioned responses (CRs). Coactivation of the FN and the IN allows for the generation of proper motor commands for CRs, but only FN output fine-tunes unconditioned responses. The vermis-FN pathway launches its signal via the contralateral ventral medullary reticular nucleus, which converges with the command from the simplex-IN pathway onto facial motor neurons. We propose that the IN pathway specifically drives CRs, whereas the FN pathway modulates the amplitudes of eyelid closure during DEC. Thus, associative sensorimotor task optimization requires synergistic modulation of different olivocerebellar modules each provide unique contributions.

[1] Department of Neuroscience, Erasmus MC, Westzeedijk 353, 3015 AA Rotterdam, the Netherlands. [2] Netherlands Institute for Neuroscience, Royal Dutch Academy of Arts & Science, 1105 BA Amsterdam, the Netherlands. ✉email: c.dezeeuw@erasmusmc.nl; z.gao@erasmusmc.nl

Sensorimotor associative behaviors allow vertebrates to convert perceptions from the environment into specific motor executions. Pavlovian delay eyeblink conditioning (DEC) is an ideal model for studying the neuronal and circuit mechanisms for associative tasks in which the motor response is precisely timed with respect to sensory input[1–3]. In this paradigm, animals are presented with a neutral conditioned stimulus (CS) followed, at a fixed interval, by an unconditioned stimulus (US) that reliably causes an unconditioned eyeblink reflex (UR). Prior to conditioning, the CS does not elicit any motor output. After conditioning, animals associate the CS with the US and generate a well-timed conditioned eyeblink response (CR) during the CS–US interval.

It is well established that both the acquisition and expression of DEC depend on the cerebellum[3–14]. The well-defined modular topographical circuitry of the cerebellum provides a unique entry for studying the contribution of specific cerebellar cortical and nuclear regions to sensorimotor tasks. Landmark studies over the past several decades have provided key evidence for the roles of the simplex lobule in the cerebellar hemisphere and its downstream target, the anterior interposed nucleus (IN), in DEC[6,8,10,11,15–24]. These regions receive both mossy fiber and climbing fiber inputs[25,26], which relay the CS and US signals, respectively[5,19,22,27,28]. Based on the activity of these inputs, various forms of synaptic and structural plasticity occur at the level of Purkinje cells (PCs) and the molecular layer interneurons during DEC learning[15,17,29,30], resulting in a prominent suppression in their simple spike activity[19,22,31]. As a consequence, IN neurons are disinhibited[17,24,32], eventually driving conditioned, but not unconditioned, eyelid closure via the downstream premotor red nucleus (RN) and facial motoneurons[15,18].

The extent to which other cerebellar modules also contribute to DEC[33] remains an open question. Vermal PCs project to the fastigial nucleus (FN), which targets vast numbers of downstream brain regions[34,35]. Indeed, FN outputs have recently been implied to play various roles in both motor and nonmotor tasks[36–41]. Anatomical studies using retrograde transneuronal tracing with rabies virus from the eyelid muscle (orbicularis oculi) have revealed prominent labeling in IN and FN, suggesting that alongside the IN module, the vermis and FN also have a potential role in controlling eyelid movements[42,43]. Recent imaging studies also revealed CS-related modulation in the vermal (lobule V and VI) PCs and granule cells[44,45]. However, physiological, functional, and anatomical evidence for the involvement of the vermis-FN pathway in controlling DEC is currently still unclear.

Here, we uncovered the involvement of a cerebellar vermis-FN pathway in the acquisition and expression of DEC, and we examined its interaction with the established simplex-IN pathway. We found that FN neurons and vermal PCs present CS-related modulations that correlate with CR amplitudes on a trial-by-trial basis. DEC-related modulation was observed in excitatory but not inhibitory FN neurons. Interestingly, unlike inhibition of IN[17], inhibiting FN output attenuated not only the acquisition and expression of CRs, but also the expression of URs. Furthermore, we observed that FN and IN modules have distinct input and output patterns and that both modules need to be coactivated to generate optimal conditioned motor commands in the downstream facial motor neurons. Viral tracing and circuit-specific perturbation revealed that the vermis-FN module controls eyelid responses via the contralateral ventral medullary reticular formation (MdV) as the main downstream hub. These data reveal how the FN module cooperates with the canonical simplex-IN-RN module in mediating DEC, and they elucidate how different cerebellar modules interact synergistically, together covering a larger functional repertoire for associative sensorimotor behavior.

## Results

**Task-related modulation of FN neurons during DEC.** Head-restrained mice were presented with a green light for 250 ms as the CS, coterminating with a 10 ms aversive periocular air puff as the US (Fig. 1a). Following 7–10 consecutive days of training, expert mice responded to the CS with a well-timed CR prior to the onset of an UR (Fig. 1b). We subsequently measured the activity of FN neurons ipsilaterally to the trained eye by recording well-isolated single units in expert mice (Fig. 1c). Diverse modulation patterns were found in FN neurons ($n = 162$ units) in response to the CS (Fig. 1d, e) and US (Supplementary Fig. 1a–c). A majority of FN neurons (86/162, 53%, Fig. 1d, e) increased their firing rates in response to the CS (termed facilitation neurons) by $64.4 \pm 8.5\%$ (mean $\pm$ s.e.m., $n = 86$ units). A minor portion of FN neurons (10%, Fig. 1d, e) decreased their firing rates in response to the CS (suppression neurons), with an average suppression of $33.2 \pm 5.5\%$ (mean $\pm$ s.e.m., $n = 16$ units). We next examined whether FN neuron modulation was specifically associated with conditioned eyelid closure or other concurrent movements that might occur during the CS–US interval. We compared the neuron activity during the CS in the trials in which mice successfully presented CRs (CR trials) with trials that did not show CRs (non-CR trials). FN modulation was significantly more prominent in CR trials than in non-CR trials (Supplementary Fig. 2a–d). Interestingly, FN activity was specifically associated with acquired eyeblink responses rather than spontaneous eyelid movements (Supplementary Fig. 2a–d). These activity features of FN neurons were comparable with those of IN neurons (Supplementary Fig. 2e–h). Both facilitation and suppression neurons had clear US-related modulation (Supplementary Fig. 1a, b). A discrete modulation feature was found particularly in neurons with both CS- and US-related facilitation ($P < 0.001$; Supplementary Fig. 1d). Approximately 37% of FN neurons did not show significant modulation during the CS (no modulation cells, $n = 60/162$ units); additionally, they presented a weaker modulation to the US (Supplementary Fig. 1e). These results suggest that the activity of FN neurons is at least partially associated with eyelid movements during DEC.

We sought to further clarify the specific relationship between FN neuron activity and the amplitudes of CRs. We analyzed the trial-by-trial correlation between the magnitudes of neuronal modulation and the amplitudes of CRs[17] (Fig. 1f–i). Out of all 86 FN neurons with CS-related facilitation, a group of neurons raised their facilitation peaks with an increase in CR peak amplitudes across trials ($P < 0.05$, linear regression, $n = 10$ units; Fig. 1f, g). In other words, the modulation amplitudes of these FN neurons were correlated with the CR peak amplitudes. Interestingly, we found a portion of facilitation neurons in which their CS-related modulation correlated negatively with the CR amplitudes ($P < 0.05$, linear regression, $n = 5$ units; Fig. 1h, i), suggesting diverse coding mechanisms for conditioned eyelid closure in FN neurons.

To analyze the temporal relationship between FN activity and CR performance, we generated a three-dimensional correlation matrix for all modulating FN neurons (see "Methods" and our previous work[17]). In short, we computed the significance of trial-by-trial correlations between FN neuronal activity and eyelid position at various epochs throughout the task. Significant correlations between FN facilitation and CR performance were found above the diagonal line of the matrix within the CS–US interval, revealing that the across-trial correlations were strongest when FN facilitation preceded eyelid closure (Fig. 1j). The peak correlation was found when FN neuron facilitation occurred 40 ms prior to the CR (Fig. 1j). In line with this, both the onset and peak times of FN facilitation were significantly earlier than the CR onset ($P < 0.001$, paired two-sided $t$ test; Fig. 1k) and peak times ($P < 0.01$, paired two-sided $t$ test; Fig. 1l). In contrast, FN neuron suppression had a minimal trial-by-trial

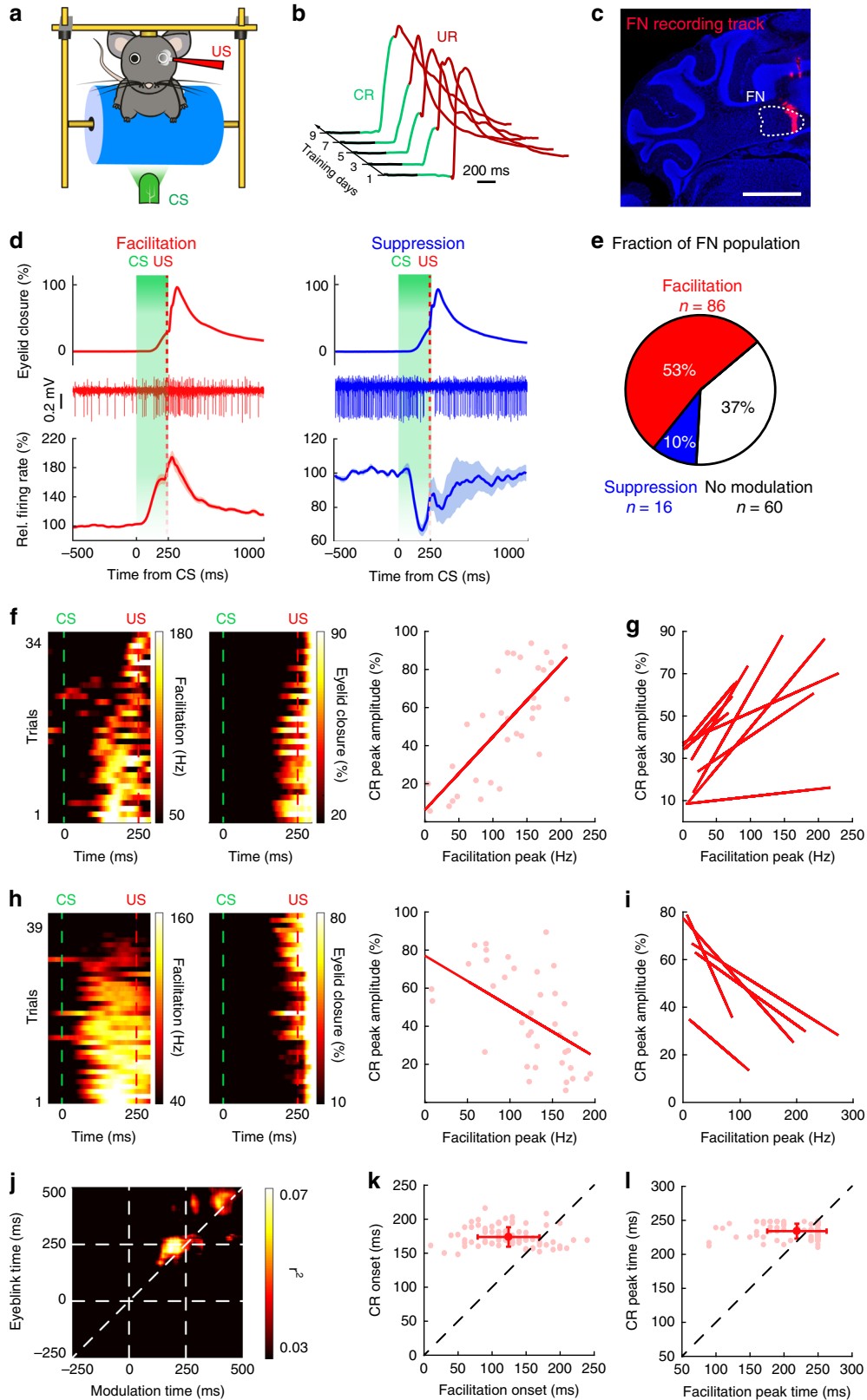

correlation with CR performance ($n = 1/60$ suppressing cells, Supplementary Fig. 3a, b). Even so, the onset and trough timings of neuronal suppression were also significantly earlier than CR onset ($P < 0.01$, paired two-sided $t$ test; Supplementary Fig. 3c) and peak time ($P < 0.001$, paired two-sided $t$ test; Supplementary Fig. 3d). Taken together, these results reveal a significant correlation between FN activity and CR amplitude, especially in the facilitation neurons.

**DEC-related modulation in FN is specific for glutamatergic neurons.** Cerebellar nuclei comprise heterogeneous groups of

**Fig. 1 Extracellular electrophysiological recordings of FN neurons during DEC. a–c** Schematic of the experimental design. **a** Head-fixed mouse is presented with a green LED light as the conditioned stimulus (CS) and a periorbital air puff as the unconditioned stimulus (US). **b** Conditioned responses (CRs, green) emerge over training days, prior to the onset of unconditioned responses (URs, red) during DEC training. **c** An example of DiI-labeled recording track in the cerebellum showing the electrophysiological recording location in the FN ($n = 7$ mice). Scale bar, 1 mm. **d** Activity of FN neurons during DEC. Top and middle rows: example traces of eyelid movement and single unit activity of FN neurons with CS-related facilitation (left) and suppression (right). Bottom row: group average of activity patterns for each modulation type ($n = 86$ for facilitation neurons; $n = 16$ for suppression neurons; mean ± s.e.m.). Green shading indicates the CS–US interval. **e** Fraction of FN neurons with different modulation types. **f** Heatmaps: an example cell with a positive correlation between neuron activity (left) and CR amplitude (right). Each row in the left heatmap represents a single trial recording and each row in the right heatmap represents the corresponding CR amplitude from the same trial. Trials are ordered from top to bottom by their peak facilitation amplitudes. Dashed lines indicate CS and US onsets. A positive trial-by-trial correlation between facilitation and CR amplitudes is shown in the right panel (linear regression model, $P = 2.74e^{-7}$) and each dot represents a single trial. **g** Summary of all facilitation cells with a positive correlation (linear regression model, $P < 0.05$, $n = 10$). **h** Same as (**f**), but for a neuron with a negative trial-by-trial correlation between facilitation and CR amplitudes (linear regression model, $P = 0.0003$). **i** Same as (**g**), but for cells with a negative correlation (linear regression model, $P < 0.05$, $n = 5$). **j** Average correlation matrix of 86 facilitation FN cells. Each epoch indicates the mean $r^2$ value of the trial-by-trial correlation between the FN neuron activity and eyelid closure at a given time point throughout the task. Most-correlated epochs (bright pixels) are located above the diagonal line and before US onset. CS and US onsets are denoted by dashed lines in both dimensions. **k** Summary of the relationship between facilitation onset and CR onset for all facilitation cells (mean ± SD, $n = 86$, paired two-sided $t$ test, $P = 1.05e^{-14}$). **l** Same as (**k**), but for the relationship of peak timings (mean ± SD, $n = 86$, paired two-sided $t$ test, $P = 0.0024$).

neurons. In general, the large excitatory neurons in the nuclei are glutamatergic and project to diverse extracerebellar regions, whereas the GABAergic inhibitory neurons project mainly to the inferior olive and/or to local FN neurons[46]. To clarify which type (s) of FN neurons are recruited in DEC, we first expressed an excitatory opsin, ChrimsonR, in either the excitatory or inhibitory neurons by stereotaxically injecting AAV9-Syn-FLEX-ChrimsonR-tdTomato into the FN of VGluT2-ires-Cre or Gad2-ires-Cre mice (Fig. 2a). ChrimsonR-expressing neurons showed robust short-latency facilitation (7.1 ± 4.2 ms for VGluT2-Cre cells, 5.5 ± 3.9 ms for Gad2-Cre cells) in response to photo-activation (Fig. 2b, c and Supplementary Fig. 4) and therefore could be "opto-tagged" as glutamatergic or GABAergic FN neurons. We identified 15 glutamatergic cells and 8 GABAergic cells (Supplementary Fig. 4) and subsequently recorded the activity of these "opto-tagged" neurons during DEC. Both CS-related facilitation (average facilitation 195.2 ± 39.6%, $n = 7$ units) and suppression responses (average suppression 67 ± 9.4%, $n = 3$ units) were found in the glutamatergic cells (Fig. 2d). In contrast, no modulation was observed in any of the GABAergic neurons, which was significantly below the chance level of detecting a modulating neuron in the FN ($P = 3.08 \times 10^{-4}$; Fig. 2e). Therefore, it is likely that glutamatergic neurons were selectively or at least predominantly recruited in DEC.

FN neurons are innervated by PCs from distinct parasagittal modules[47,48]. We sought to identify the cerebellar cortical regions that project to DEC-related FN neurons. We focused on FN neurons with CS-related facilitation, as these neurons were most prevalent and had significant trial-by-trial correlations with CR amplitudes (Fig. 1f–i). We performed single glass pipette juxtacellular recordings to identify neurons with CS-related facilitation in the FN (Fig. 2f). Subsequently, cholera toxin b-subunit (CTB) in the recording pipette was injected in the vicinity of the identified region (Fig. 2g, left panel, see "Methods"). Retrogradely labeled PCs were observed exclusively in the cerebellar vermis (Fig. 2g, right panel). Overall, CTB-labeled PCs were found in restricted parasagittal areas of vermal lobules IV to VIII, centered approximately 300 μm from the midline (Fig. 2h), corresponding to the b zone which receives its climbing fiber input from the caudal dorsal accessory olive (DAO)[49]. No labeled PCs were detected in the canonical DEC-related cerebellar region, i.e., the simplex lobule. Hence, DEC-related FN neurons are likely to receive task-related information from PCs in specific cerebellar vermal regions.

**DEC-related simple spike and complex spike modulation of vermal PCs.** Associative conditioning depends on the cerebellar cortex[10,13,19,22,28,32,50]. Conditioned PCs in the simplex lobule present a delayed simple spike pause in response to the CS, which is considered crucial for the acquisition and expression of CRs[10,19,28,50]. Here, we asked what information vermal PCs encode during DEC and whether they share a modality with simplex PCs. To test the involvement of vermal PCs during DEC, we recorded PC activity from vermal lobules IV to VIII ipsilaterally to the trained eye, which we identified as the task-relevant regions for DEC (Figs. 2h, 3a). Well-isolated PCs were identifiable with their stereotypical simple spike and complex spike waveforms (Fig. 3b). A majority of the vermal PCs modulated their activity during the CS–US interval (Fig. 3c, d). Specifically, one-third of the PCs decreased their simple spike firing rates during the CS–US interval (SS suppression, firing rate decreased 18.9 ± 2.9%, $n = 23/62$ units; Fig. 3c, d), similar to the PC activity pattern in the simplex lobule during DEC[19,28,31]. Another group of PCs increased their simple spike firing rates in response to the CS (SS facilitation, firing rate increased 29.5 ± 4.5%, $n = 26/62$ units; Fig. 3c). Compared to a clear CS-related modulation during CR trials, cells exhibiting either SS suppression or facilitation had weaker modulation in the non-CR trials and minimal activity changes in response to a spontaneous blink (Supplementary Fig. 5), further supporting the task specificity of their modulation. US-related simple spike modulations were identified in both SS suppression and SS facilitation PCs (Supplementary Fig. 6a, b). However, a significant correlation between the CS- and US-related modulation amplitude was only found in the PCs exhibiting SS suppression (Supplementary Fig. 6c).

Given the significant trial-by-trial correlation between FN firing rates and CR amplitudes (Fig. 1f–i), we next analyzed the relationship between simple spike modulation and CR peak amplitude on a trial-by-trial basis. Indeed, a positive correlation was found in a subgroup of PCs exhibiting SS suppression (linear regression, $P < 0.05$, $n = 8/23$ units; Fig. 3e, f). The temporal relation between SS suppression and CR amplitude was further analyzed with a correlation matrix (see Methods), showing that the strongest correlation occurred 40 ms prior to the US (Fig. 3g). For PCs with SS facilitation, the activity hardly correlated with CR amplitudes on a trial-by-trial basis, yielding only one cell with a significant correlation (linear regression, $P < 0.05$; Supplementary Fig. 7a, b). We found that both the onset and peak of CS-related modulation occurred earlier than the initiation and peak timing of CR, in PCs exhibiting either SS facilitation or suppression (Fig. 3h and Supplementary Fig. 7c). Taken together, these results reveal a cerebellar cortical module for DEC and suggest that SS suppression in vermal PCs in turn might facilitate FN neurons and modulate the timing and amplitude of eyelid closure during DEC.

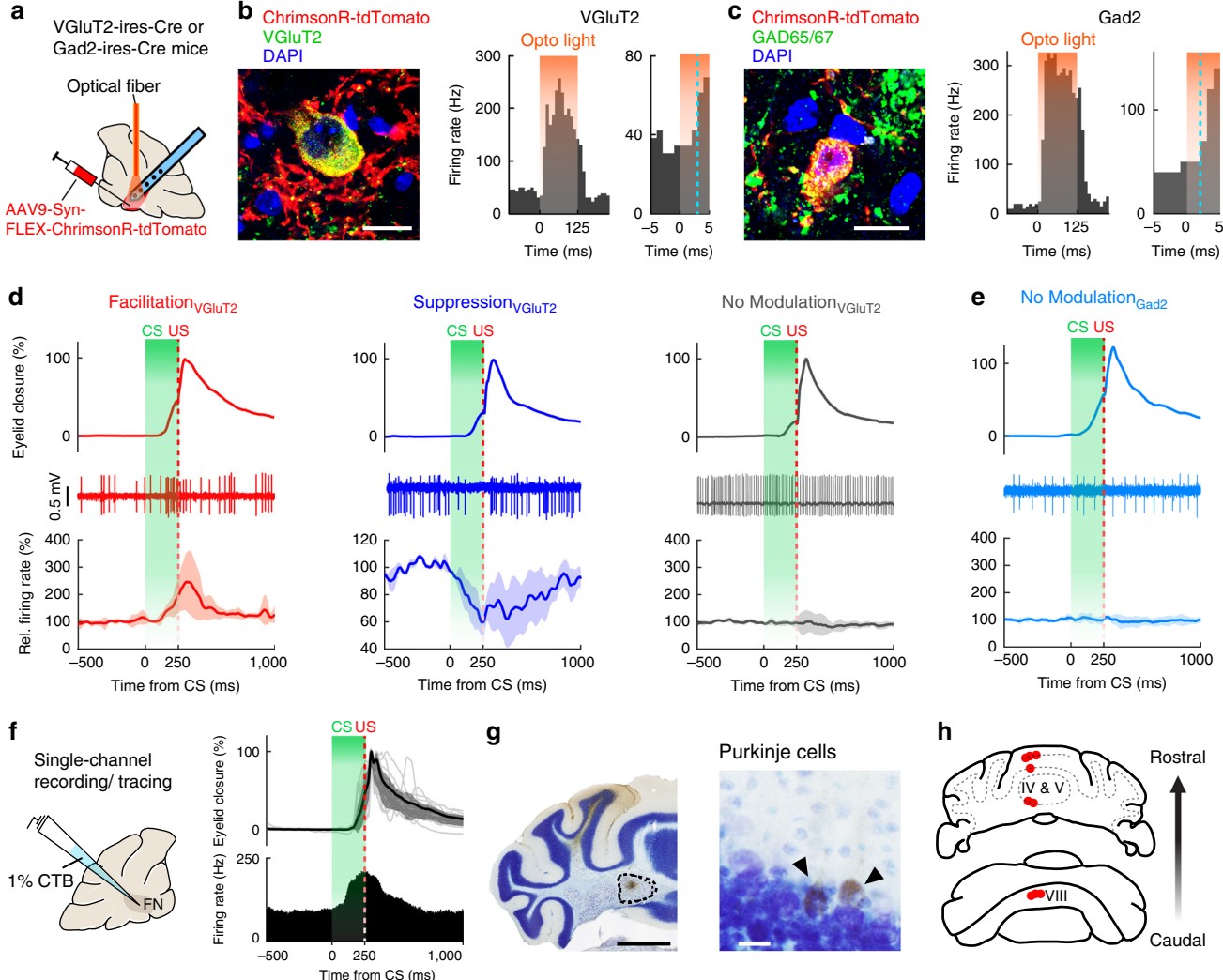

**Fig. 2 Task-related modulation in excitatory FN neurons and the identification of DEC-related vermal regions. a** Schematics showing viral injection, optical fiber implantation and multichannel recording in the FN of the VGluT2-ires-Cre mice ($n = 5$) or the Gad2-ires-Cre mice ($n = 8$). **b**, Expression of Cre-dependent ChrimsonR in VGluT2-positive FN neurons (left), showing a short-latency response to 595 nm light (orange shading, right). The blue dashed line indicates the timing at which the firing rate exceeds three SDs of the baseline frequency within 20 ms after the light. Scale bar, 10 μm. **c** Same as (**b**), but for the Gad2-positive neurons. **d** Task-related modulation of VGluT2-positive neurons. Neurons are categorized based on their CS-related modulations. Top and middle rows: example eyelid movement and spike traces of individual cells. Bottom row: average firing rate of neurons with CS-related facilitation (left, $n = 7$), suppression (middle, $n = 3$) and no modulation (right, $n = 5$); traces are plotted as mean ± s.e.m. **e** Same as (**d**), but for the Gad2-positive neurons, showing no CS-related modulation ($n = 8$). **f** Left: experimental design for FN neuron recording and CTB tracing by using a single glass capillary. Right: a representative neuron showing CS- and US-related facilitation (overlaying eyelid closure, mean ± SD, $n = 21$ trials, upper right; PSTH, lower right) during the CS–US interval. **g**, **h** Iontophoresis of CTB localized to the recording site (**g**, left, scale bar, 1 mm) and retrogradely labeled Purkinje cells (**g**, right, scale bar, 20 μm) in the parasagittal vermis regions (**h**) ($n = 5$ mice).

CS-related complex spikes ($Cpx_{CS}$) in the PCs of simplex lobule encode crucial instructive signals for CR acquisition and expression[19,22,28], and short-latency US-related complex spikes ($Cpx_{US}$) are considered to carry the canonical IO signal. We next opted to address whether PCs in the vermal DEC region also have specific complex spike firing patterns in response to the CS and US. In total, 29 vermal PCs increased their complex spike firing rate following CS (Wilcoxon rank-sum test, $P < 0.05$; Fig. 4a). The majority of these PCs, 23/29 neurons presented short-latency complex spikes in response to the US ($26.9 \pm 2.6$ ms after US, Supplementary Fig. 8a), $41.9 \pm 2.8$ ms before the UR peak (mean ± s.e.m., Supplementary Fig. 8d). Short-latency $Cpx_{US}$ were recorded in both SS suppression PCs ($n = 10/23$, Supplementary Fig. 8b) and SS facilitation PCs ($n = 15/26$, Supplementary Fig. 8b). The other 6 neurons with $Cpx_{CS}$ were not significantly

modulated following US (Supplementary Fig. 8c). Similar to previous findings in simplex lobule PCs[22], the modulation amplitude of vermal $Cpx_{CS}$ correlated with $Cpx_{US}$ in trained mice (Supplementary Fig. 8e).

Given the different properties of the CS-related simple spike modulations (Fig. 3 and Supplementary Fig. 7), it is possible that the $Cpx_{CS}$ of PCs exhibiting SS suppression carry information for CRs that differs from that of PCs exhibiting SS facilitation. We therefore investigated the relation between the $Cpx_{CS}$ activity and their corresponding CR performance in terms of timing and amplitude. PCs with $Cpx_{CS}$ were categorized based on their simple spike activity during CR (Fig. 4a). Interestingly, whereas $Cpx_{CS}$ were prominently detected in both PCs demonstrating SS suppression and PCs demonstrating facilitation ($n = 12$ for each), only the $Cpx_{CS}$ of PCs exhibiting SS suppression had an earlier

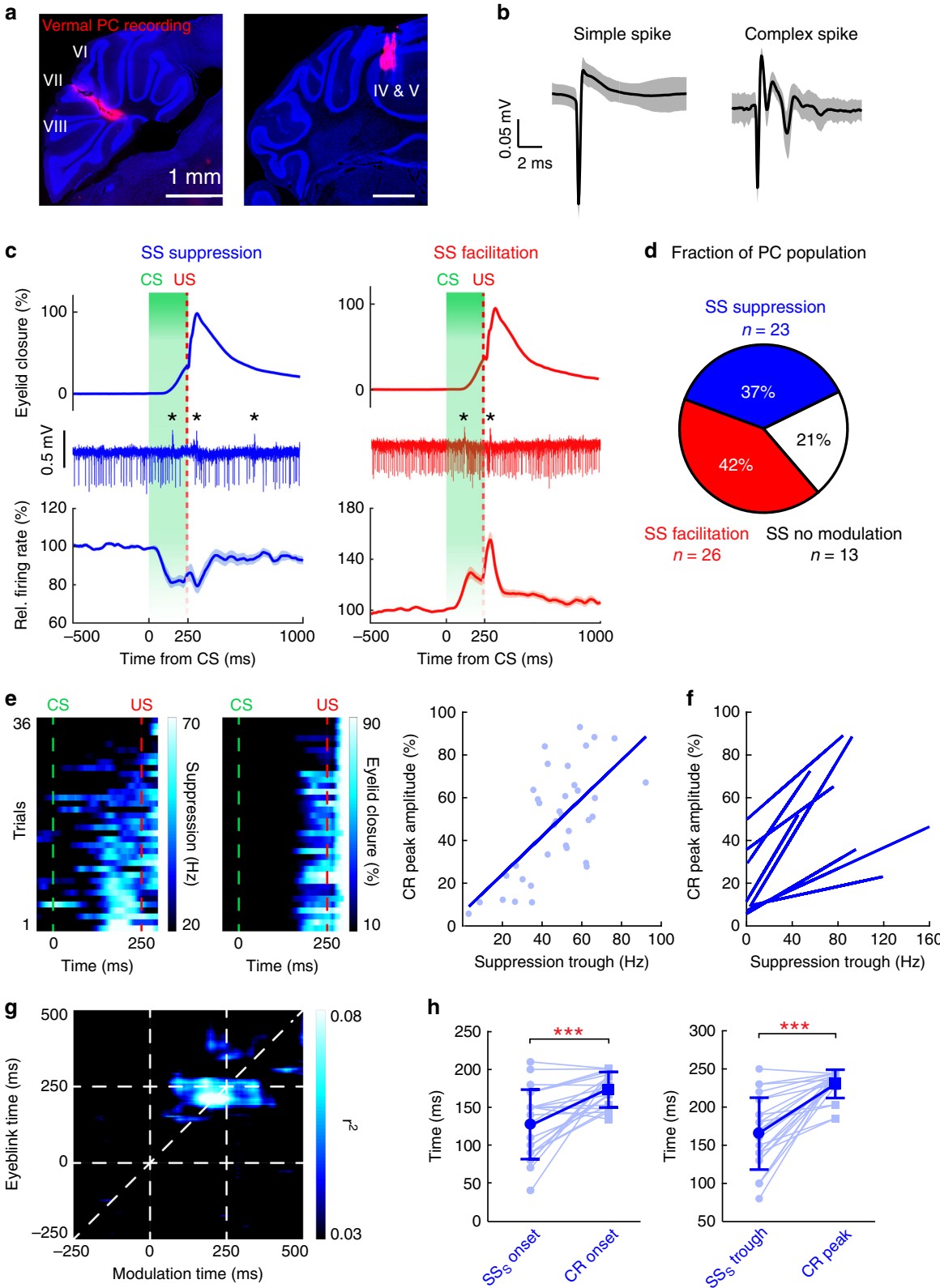

onset timing than eyelid closure (paired two-sided $t$ test, $P < 0.05$; Fig. 4b). For each individual neuron, the $Cpx_{CS}$ demonstrated a consistent latency despite the variable initiation of CRs (Fig. 4c, d). To examine the relation between $Cpx_{CS}$ and CR amplitudes, we divided the trials based on the occurrence of $Cpx_{CS}$. Mice had larger CR amplitudes when $Cpx_{CS}$ occurred within the 50–250 ms window after CS delivery (paired two-sided $t$ test, $P < 0.01$; Fig. 4e,

f); this correlation was only found in the PCs showing SS suppression (Fig. 4e–g). Hence, our results not only uncover the relation between SS suppression of vermal PCs and behavior, but also highlight the role of vermal PC $Cpx_{CS}$ in DEC.

**Shared and distinct contributions of FN and IN outputs to DEC learning and behavior.** Our results unequivocally

**Fig. 3 Task-related simple spike modulation in vermal PCs. a** Representative DiI-labeled recording tracks in cerebellar vermal regions (lobules IV-VII). Scale bars, 1 mm. Experiments were performed with 17 mice. **b** Representative waveforms (mean ± coefficient of variation) of simple spikes and complex spikes from a single PC. **c** CS-related simple spike modulation in vermal PCs. Top and middle rows indicate example eyelid closure and spike traces of individual PCs (* indicates complex spikes); bottom: group average of simple spike activity from PCs of each modulation type (blue: PCs with simple spike suppression, $n = 23$; red: PCs with simple spike facilitation, $n = 26/62$), traces are plotted as mean ± s.e.m. **d** Fraction of PC population with simple spike modulations. **e** Example PC with a significant correlation between the simple spike suppression (left heatmap) and the CR peak amplitudes (right heatmap) over trials. Each row represents a single trial, ordered from bottom to top based on the magnitude of the simple spike suppression. The correlation of this cell is shown in the right panel (linear regression model, $P = 1.24e^{-5}$), and each dot represents a single trial. **f** Summary of all PCs showing a significant trial-by-trial correlation between simple spike suppression and CR peak amplitude (linear regression model, $P < 0.05$, $n = 8$). **g** Average correlation matrix of 23 suppressed cells. Most-correlated epochs (bright pixels) are distributed across the diagonal line and before US delivery. CS and US onsets are denoted with dashed lines in both dimensions. **h** Comparison of the timing of simple spike suppression and behavior. Simple spike suppression precedes the CR both in onset (left, mean ± SD, $n = 23$, paired two-sided $t$ test, ***$P = 0.00013$) and peak timing (right, mean ± SD, $n = 23$, paired two-sided $t$ test, ***$P = 1.94e^{-5}$).

demonstrate task-related modulation in the vermis-FN module. Since the majority of FN neurons increased their firing rates during the CS–US interval (Fig. 1d–j) and task-related modulation was found solely in excitatory neurons (Fig. 2d), we examined the necessity of FN output for controlling DEC by pharmacologically inhibiting FN neuron activity with the GABA$_A$ receptor agonist muscimol[51] (Fig. 5a). Precise muscimol injections targeting the FN ipsilaterally to the trained eye, largely abolished CRs in conditioned mice (Fig. 5b, c). Interestingly, inhibiting FN activity also suppressed eyelid closure in response to the US by reducing the UR peak amplitudes over 40% (Fig. 5b, c). Both CR and UR performance recovered fully after washing out the muscimol (Fig. 5b, c). These results suggest the functional necessity of FN neuron activity for CR and UR performance during DEC.

To pinpoint whether FN modulation, specifically during the CS–US interval, is essential for CRs and URs, we transiently suppressed FN activity during the CS–US interval by photo-activating the axon terminals of ChR2-expressing PCs in L7Cre-Ai27 mice (Fig. 5d). Light intensity (470 nm wavelength, <1.5 mW) was carefully adjusted so that no obvious aversive behavior, locomotion impairment, or suppression of the neighboring IN neurons was observed under this condition (see control data in our previous work[38]). Similar to the effects of long-term muscimol inhibition, transient suppression of FN output within the CS–US interval sufficiently impaired the CR, as well as UR performance (Fig. 5e, f). In contrast, optogenetic inhibition of IN output during the CS–US interval specifically suppressed the CR, leaving the UR intact (Fig. 5g–i). To further exclude the possibility that inhibiting FN activity could impair IN facilitation during DEC, we recorded task-related activity in the IN while optogenetically inhibiting the FN during the CS–US interval (Supplementary Fig. 9a). Despite the significant suppression of behavior, inhibiting FN output enhanced task-related modulation in IN neurons (Supplementary Fig. 9b, c). Therefore, it is unlikely that inhibiting FN activity affects eyelid closure due to its effect on IN facilitation.

Our pharmacological and optogenetic manipulations had robust effects on CR performance in trained mice, but it could still be the case that acutely shutting down the FN output causes transient disruption of downstream target regions, thereby affecting CR performance only temporarily. To better demonstrate the enduring necessity of FN in CR expression, we chronically ablated the ipsilateral FN using photolesions in well-trained mice (Fig. 5j, k, see "Methods") and tested their CR performance for three consecutive post-lesion days (Fig. 5l, m). Chronic FN lesions significantly impaired CR performance: both the CR-trial probability and the CR amplitude were smaller in FN lesion mice compared to the control mice that underwent a sham operation (Fig. 5l, m). These CR impairments were evident

throughout three post-lesion days without clear recovery (Fig. 5n, o). Hence, ablating FN had long-lasting effects on CR performance. Taken together, our results suggest that FN and IN outputs are both essential for CR expression; the different effects of FN/IN inhibition on URs indicate distinct mechanisms of these two cerebellar modules in mediating eyelid movement during DEC. Previous studies have established a crucial role for the IN in driving eyelid closure during DEC learning and behavior[8,17,52–56]. To further illustrate the functional distinction between FN and IN pathways, we examined whether FN output could also directly drive eyelid closure. We electrically activated either the IN or the FN in naive mice (Supplementary Fig. 10a, b). In line with previous findings[8], eyelid closure was robustly elicited by electrical activation of the IN with graded current intensities (Supplementary Fig. 10b, c). However, the same electrical stimulation conditions in the FN region inadequately drove eyelid closure (Supplementary Fig. 10a, c), supporting that FN facilitation is not the direct driver for eyelid closure during DEC, but a muscle tone modulator that is expressed during both the CR and UR. Therefore, our results reveal the functional similarity and difference of two cerebellar modules in controlling eyelid closure and highlight the unique role of FN in modulating, but not driving, CR and UR performance.

The experiments described above indicate that the FN module is required for the expression of the CRs and URs following acquisition, yet they do not directly demonstrate its role during the acquisition itself. We next tested whether the vermis-FN module was also required for the acquisition of CR by using chemogenetic (long-term) and optogenetic (timing-specific) suppression of FN outputs during DEC training. Inhibitory DREADDs were virally expressed in the FN unilateral to the eye that received DEC training, and tdTomato was expressed in control mice (Fig. 6a). The activity of DREADD-expressing FN neurons in awake mice was significantly decreased after *i.p.* clozapine-N-oxide (CNO) administration (Fig. 6b, c). Therefore, we injected CNO daily in both groups, 15–20 min prior to DEC training for 10 days in a row. CR acquisition (CR amplitude and probability) in the DREADD-inhibition group was significantly impaired compared to that of the control group (maximum likelihood estimation, $P < 0.001$; Fig. 6d, e). After these 10 training days, we tested the acquisition outcome with the CNO injection omitted on day 11. Compared to the control group, the DREADD-expressing mice showed a significantly smaller CR amplitude and CR probability (Fig. 6f). To control for potential side effects of chronic DREADD expression or CNO administration on DEC training, we next optogenetically activated vermal PCs in L7Cre-Ai27 animals, which allowed us to transiently suppress FN activity, specifically during the training epoch. Optic light (470 nm wavelength, <1.5 mW, as in Fig. 5d–i) was given to the ipsilateral FN of the training eyes in both L7Cre-Ai27 and

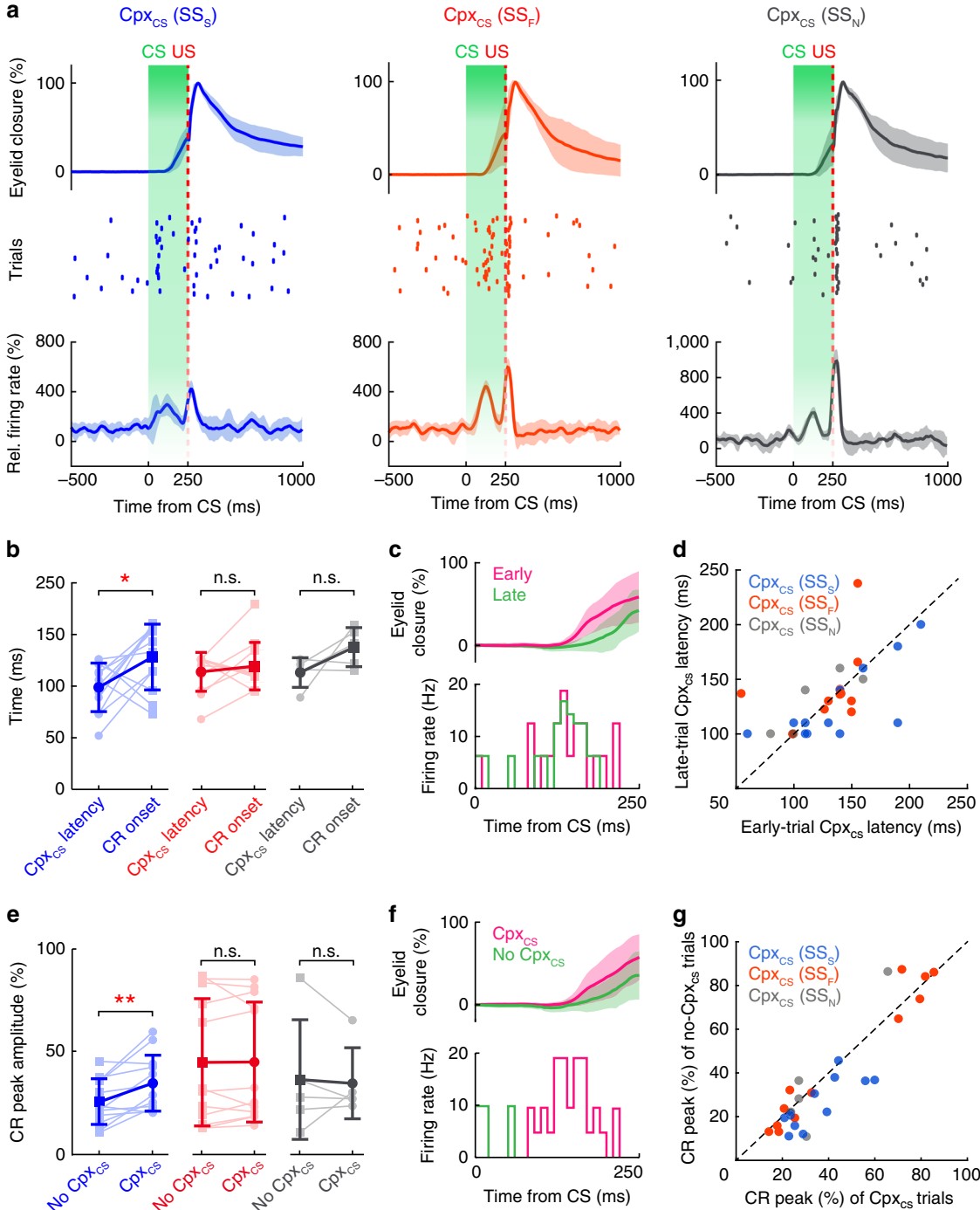

**Fig. 4 Purkinje cell complex spikes encode CR-related information. a** Complex spike modulation during DEC. PCs with CS-related complex spikes ($Cpx_{CS}$) are color-coded based on their simple spike (SS) modalities: suppression ($Cpx_{CS}(SS_S)$, blue), facilitation ($Cpx_{CS}(SS_F)$, red) and no modulation ($Cpx_{CS}(SS_N)$, gray). Top row: summary of eyelid responses (left to right: $n = 30, 32, 41$ trials, mean ± SD). Middle row: example complex spike activity (raster plots of spike events) during DEC, and bottom row shows average $Cpx_{CS}$ activity of each PC population (left to right: $n = 12$, $n = 12$, $n = 5$ neurons, mean ± s.e.m.). **b** Comparison between the timing of $Cpx_{CS}$ ($Cpx_{CS}$ latency) and the CR onset. Only PCs with simple spike suppression showed an earlier occurrence of $Cpx_{CS}$ than CR onset (mean ± SD, paired two-sided $t$ test, left to right: $n = 12, 12,$ and 5, $P = 0.04, 0.51,$ and 0.14). **c** Comparison of $Cpx_{CS}$ latency in trials divided into early ($n = 16$ trials, 147.4 ± 23.6 ms, mean ± SD) and late trials ($n = 16$ trials, 196.8 ± 21.2 ms, mean ± SD) based on CR onset. Example recording of $Cpx_{CS}$ during the CS–US interval (firing rate PSTH, **c**, bottom) in the early and late CR trials (**c**, top). **d** Population summary showing no difference in $Cpx_{CS}$ latency between early and late trials in any category of PCs (paired two-sided $t$ test, $P = 0.39$). **e** Comparison of CR peak amplitudes in trials with and without $Cpx_{CS}$. The occurrence of $Cpx_{CS}$ in the PCs with simple spike suppression predicts a larger CR amplitude (mean ± SD, paired two-sided $t$ test, left to right: $n = 12, 12,$ and 5, $P = 0.005, 0.94,$ and 0.80). **f** Example traces of CRs (top, $n = 21$ trials for pink trace, $n = 10$ trials for green trace, mean ± SD) with or without $Cpx_{CS}$ (firing rate PSTH, bottom). $Cpx_{CS}$ is defined as the complex spikes that occur within 50–250 ms following CS onset. Correlation of $Cpx_{CS}$ occurrence and CR peak amplitude for three categories of Purkinje cells is summarized in (**g**). PCs with simple spike suppression ($Cpx_{CS}(SS_S)$, blue) reside below the diagonal line.

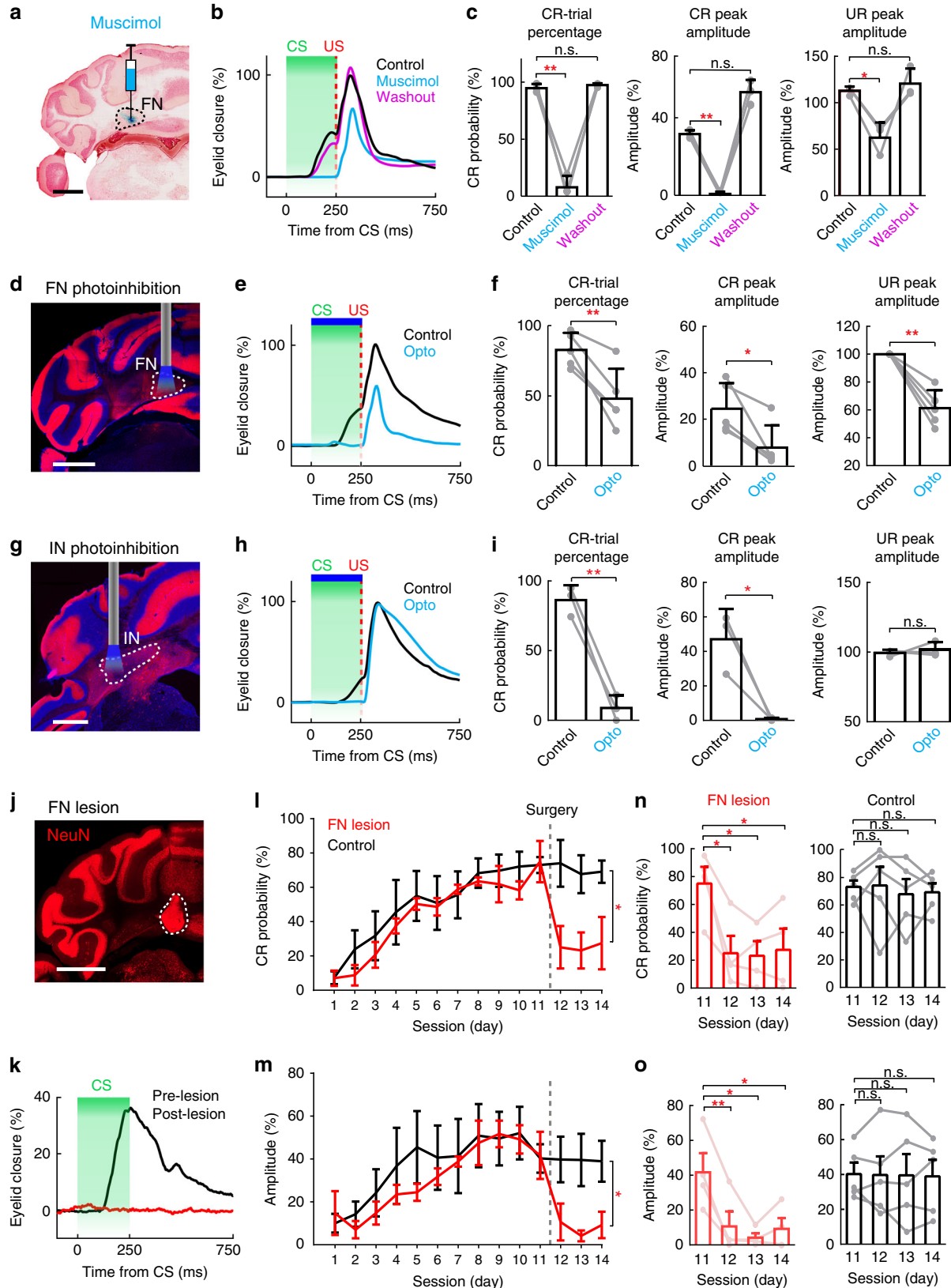

control (wild-type) mice. Similar to the DREADD experiment, CR acquisition was significantly impaired during the 10 training days (maximum likelihood estimation, $P < 0.001$; Fig. 6g, h) and CR performance was significantly worse in that the amplitudes were smaller and the probabilities were lower on test day 11 (optic light omitted; Fig. 6i). Therefore, the vermis-FN module is

crucial not only for mediating CR and UR expression with a proper muscle tone in conditioned mice, but also for optimal CR acquisition during the DEC learning process.

**Synergistic activation of IN and FN pathways is permissive for movements.** To determine how two cerebellar modules

**Fig. 5 Effects of transient and chronic FN perturbation on the expression of DEC. a** Example injection site of muscimol and alcian blue in the FN ipsilateral to the trained eye. Scale bar, 1 mm. **b** CR and UR performance of a mouse following muscimol inhibition of the FN. Average traces of eyelid movement, in control (black), by muscimol inhibition (cyan), and after washout (magenta) sessions from the same mouse. **c** Summary of the effects of muscimol inhibition on CR and UR performances ($n = 3$ mice, mean ± SD, paired two-sided $t$ test, *$P < 0.05$, **$P < 0.01$). **d** Example cerebellar section showing exclusive expression of ChR2-tdTomato in the PCs of L7Cre-Ai27 mice. An optic fiber was implanted above the FN ipsilateral to the trained eye. Scale bar, 1 mm. **e**, **f** Same as (**b**, **c**), but for the optogenetic perturbation of FN neurons during the CS-US interval (indicated in blue bar). Both CRs and URs were suppressed ($n = 5$ mice, mean ± SD, paired two-sided $t$ test, *$P < 0.05$, **$P < 0.01$). **g–i** Same as (**d–f**), but for optogenetic perturbation of the simplex lobule-IN module. CRs, but not URs, were suppressed ($n = 3$ mice, mean ± SD, paired two-sided $t$ test, *$P < 0.05$, **$P < 0.01$). **j** Example image of cerebellar section after laser photolesion. Dashed contour highlights the strong autofluorescence from FN lesion site ($n = 4$ mice). Scale bar, 1 mm. **k** Representative CR traces from a trained mouse before (black) and after FN lesion (red). **l** Summary of CR-trial probabilities in the control ($n = 5$, mean ± s.e.m., black trace) and FN lesion groups ($n = 4$, mean ± s.e.m., red trace). CR-trial probability in lesion animals was lower than that of the control group (two-way repeated measures ANOVA, *$P < 0.05$). Dashed line indicates the time point for the photolesion. **m** Same as (**l**), but for the comparison of CR amplitudes in two groups. **n** Comparison of the CR-trial probability between the pre-lesion session (day 11) and 3 post-lesion sessions (days 12–14), in lesion group (red, $n = 4$, two-way repeated measures ANOVA, *$P < 0.05$) and control group (black, $n = 5$, paired two-sided $t$ test, $P > 0.05$). Dots and lines indicate performance of different mice, mean ± s.e.m. **o** Same as (**n**), but for the comparison of CR amplitudes before and after lesion in the lesion group ($n = 4$, paired two-sided $t$ test, *$P < 0.05$, **$P < 0.01$) and control group ($n = 5$, $P > 0.05$). See the exact $P$ values for each comparison in the Source Data file.

synergistically contribute to eyelid closure during DEC and to clarify the integration of these cerebellar outputs in generating eyelid motor commands, we recorded the motor neurons of eyelid muscles during DEC, while photoinhibiting either IN or FN output in the same animal (Fig. 7a). The eyelid muscle (*orbicularis oculi*) is controlled by motor neurons of the facial nucleus (7N)[42,43,57], which can be readily identified by their anatomical location (see Supplementary Table 1) and activity patterns during spontaneous as well as DEC-induced eyelid closures (Fig. 7b). When we inhibited either IN or FN output during the CS–US interval, CS-related modulation in the 7N neurons was consistently significantly reduced ($n = 19$ units; Fig. 7c–f). Moreover, in 7 out of the 19 cells, these manipulations even suppressed the 7N neuron firing rates below the baseline CS response levels (Fig. 7f, upper panel). The average decrease in 7N neuron activity following CS was comparable for IN- and FN-inhibition trials, resulting in a decrease to 6.7 ± 19.9% and −13.8 ± 24.9% of the baseline, respectively. The effects of inhibiting the IN or the FN were supralinear in that the arithmetic sum of reduction in 7N activity by FN plus IN inhibition was 207% of the average CS-related modulation amplitude in control trials ($P < 0.03$). In contrast, only inhibiting FN output, but not IN output, suppressed the US-related modulation of 7N neurons to 29.5 ± 20.1% (Fig. 7f, lower panel), which was consistent with the behavioral outcome. Thus, our data indicate that both IN and FN outputs are essential for 7N motor neuron modulation during DEC. This suggests that synergistic activation of the IN and FN pathways is permissive for generating motor commands for CRs; whereas only the FN, not the IN, contributes to the activation of 7N neurons during URs.

**FN-MdV and IN-RN pathways converge onto the 7N and regulate DEC.** Cerebellar circuits are organized in repetitive parasagittal modules[47,58]. Previous studies have unequivocally established a key cerebellar pathway for DEC, in which IN neurons innervate the premotor neurons in the RN that subsequently excite the 7N motor neurons responsible for eyelid movements[42,43,57]. Since we uncovered an additional cerebellar pathway for DEC, i.e. the vermis-FN pathway and this pathway contributes synergistically with the simplex lobule-IN pathway, we sought to clarify the anatomical organization of the vermis-FN pathway for DEC. Therefore, we combined anterograde tracing of AAV1-CB7-RFP from the FN with retrograde tracing of AAVretro-CAG-GFP in the ipsilateral 7N (Fig. 8a) and surveyed the extracerebellar regions that link FN output to the 7N. Unlike the dense innervation from the IN, the contralateral RN received very sparse projections from the FN (Fig. 8b and Supplementary

Fig. 11a, d, e), suggesting that DEC-related FN neurons are unlikely to control eyelid closure via the RN. However, we observed extensive overlaps of FN axons with retrogradely labeled neurons from the 7N in the contralateral ventral medullary reticular nucleus (MdV) (Fig. 8c, d), which received minimal projections from the IN (Supplementary Fig. 11b, c). Higher magnification images revealed that FN axon terminals formed close dendritic and somatic appositions with 7N-projecting MdV neurons (Fig. 8e). These anatomical findings suggest that the cerebellar vermal module controls 7N motor neurons via a discrete FN-MdV pathway.

To examine whether the FN-MdV pathway may indeed mediate DEC, we manipulated this pathway by injecting Cre-dependent AAV1-hSyn-FLEX-SIO-StGtACR2 in FN and retro-grade AAVretro-hSyn-Cre-BFP in the contralateral MdV (Fig. 9a). The inhibitory opsin StGtACR2[59] was expressed exclusively in the somas of MdV-projecting FN neurons (Fig. 9b), which allowed us (1) to identify these neurons by optogenetics and to further examine their activity during DEC (Fig. 9c, d); (2) to examine the effects of specifically perturbing the FN-MdV pathway on CR and UR performance (Fig. 9e, f). Among the 15 identified "opto-tagged" MdV-projecting FN neurons (Fig. 9c), 40% showed CS-related modulation (Fig. 8d), supporting the involvement of the FN-MdV pathway in DEC. In trained animals, both CR probability and amplitudes were significantly suppressed when we photo-inhibited the FN-MdV pathway (paired two-sided $t$ test, $P < 0.01$; Fig. 9e), which was consistent with our results of pharmacological inhibition of FN (Fig. 5a–c) and optogenetic perturbation of the vermis-FN module (Fig. 5d–f). Likewise, UR amplitudes were also significantly impaired by inhibiting the FN-MdV pathway (paired two-sided $t$ test, $P < 0.01$; Fig. 9f). Thus, the FN-MdV pathway differed not only anatomically, but also functionally from the IN-RN pathway in that it is crucial for modulating both the CR and the UR during DEC.

Taken together, our results uncover a vermis-FN-MdV pathway for the associative DEC and shed light on the potential convergence and synergy in controlling downstream motor neurons to fine-tune eyelid movements (Fig. 9g). Therefore, our study provides new insights into the anatomical and physiological framework for studying cerebellar multimodular interactions during associative motor learning.

## Discussion
In this study, we provide evidence for the involvement of a FN-MdV pathway in associative learning and behavior, showing how it may interact and cooperate with the canonical IN-RN pathway

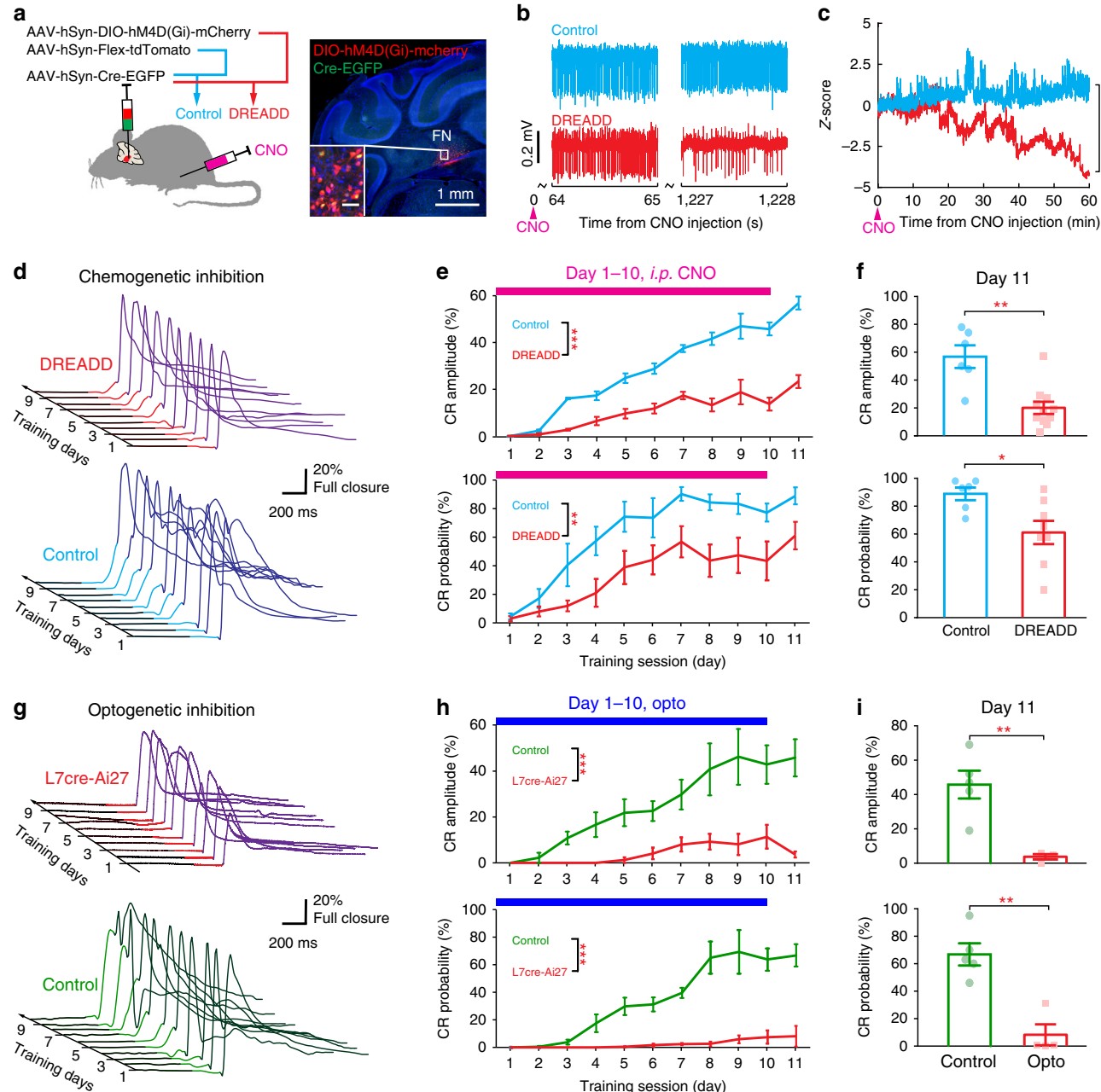

**Fig. 6 Effects of Inhibiting the Ipsilateral FN on DEC Acquisition. a** Experimental design for chemogenetic inhibition of the FN during DEC training (left). Inhibitory DREADD-hM4D (Gi) was expressed in FN ipsilateral to the trained eye (right, $n = 6$ mice) and tdTomato was expressed in control mice ($n = 8$). Both control and DREADD mice received a i.p. CNO injection 15–20 min prior to training. Scale bar for the inserted image is 50 μm. **b** Representative FN neuron responses from control and DREADD-expressing animals at early (left) and late (right) stages of recording, following the CNO injection. **c** Comparison of neuron activity over time after the CNO injection in control ($n = 37$ neurons) and DREADD-expressing ($n = 19$ neurons) mice (two-way repeated measures ANOVA, ***$P < 0.001$). **d** Progression of CR traces during DEC training in a representative DREADD mouse (CS–US interval shown in red) and a control mouse (CS–US interval shown in cyan). **e** Comparison of the CR acquisition during training (1–10 days), illustrated in the CR peak amplitude (upper) and CR-trial probability (lower), in control ($n = 8$ mice, mean ± s.e.m.) and DREADD-expressing mice ($n = 6$ mice, mean ± s.e.m., maximum likelihood estimation (two-sided), **$P < 0.01$, ***$P < 0.001$). **f** Comparison of CR performance on the 11th day with the CNO injection omitted (mean ± s.e.m., two-sample $t$ test (two-sided), *$P < 0.05$, **$P < 0.01$). **g–i** Same as in (**d–f**), but for DEC training in mice with optogenetic perturbation. CR acquisition is suppressed in L7Cre-Ai27 mice ($n = 4$ mice, mean ± s.e.m.) compared to the acquisition observed in the control group ($n = 5$ mice, mean ± s.e.m., maximum likelihood estimation (two-sided), ***$P < 0.001$). Comparison of CR performance on the 11th day with opto-inhibition omitted (mean ± s.e.m., two-sample $t$ test (two-sided), **$P < 0.01$). See the exact $P$ values for each comparison in the Source Data file.

during DEC. We found well-timed modulations in a group of excitatory FN neurons in response to the CS and US, sufficiently allowing the prediction of the CR amplitude on a trial-by-trial basis. Consistent with the DEC-related modulation in the FN, its upstream vermal PCs showed modulations of both their simplex

spikes and complex spikes in relation to both the CS and US. Reversible manipulations of the vermis-FN module revealed the functional necessity of this pathway for CR acquisition and expression, as well as UR performance. Using anatomical tracing, we demonstrated that the FN-MdV pathway directly projects to

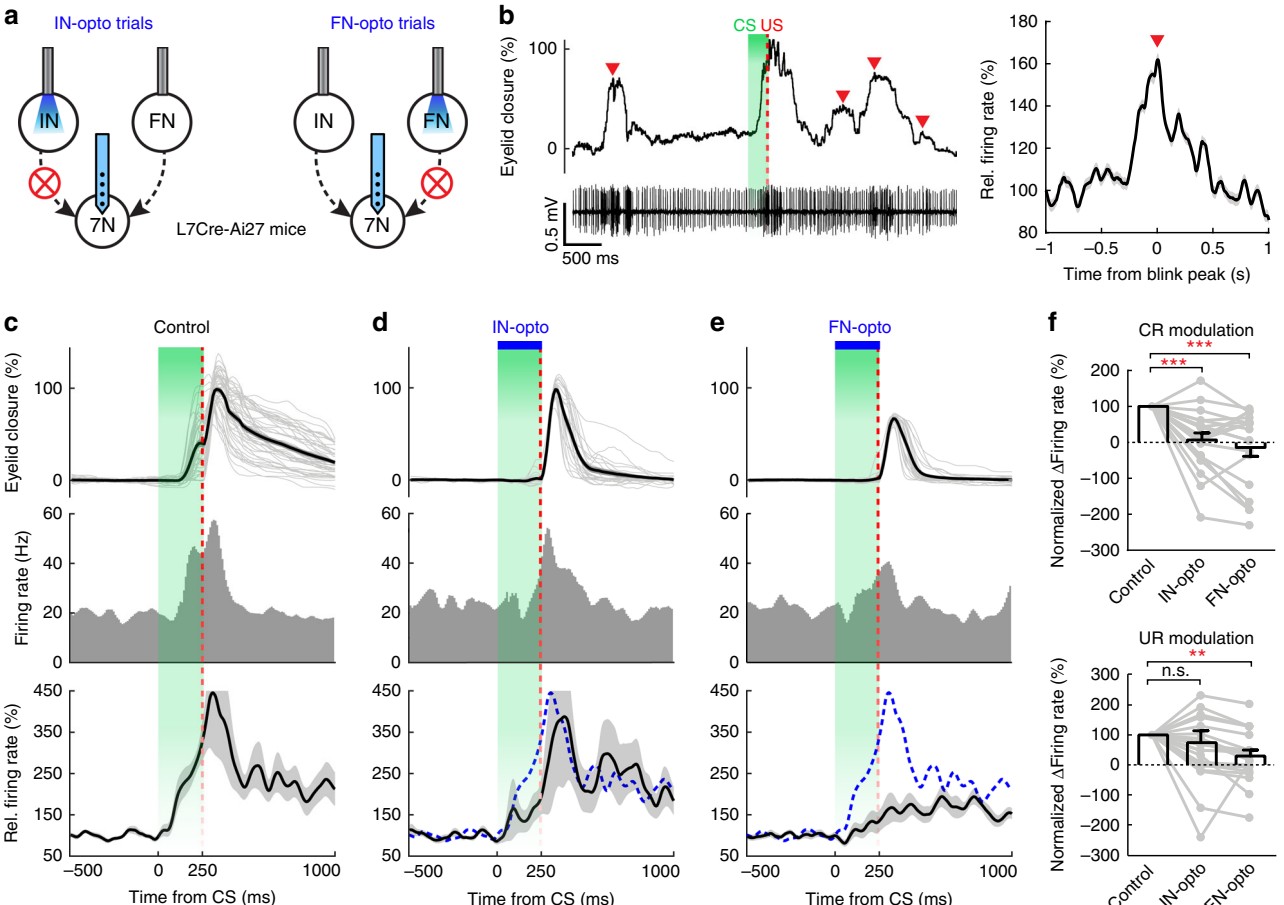

**Fig. 7 Integration of FN and IN signals in 7N motor neurons synergistically controls associative behavior. a** Experimental design of 7N neuron recording during DEC from L7cre-Ai27 mice with either IN (left) or FN (right) inhibition. Arrow-headed lines indicate outputs from the IN and the FN to 7N motor neurons. **b** Putative 7N motor neuron activity during DEC and spontaneous eyelid movements. Left: example recording of eyelid movement (upper) and a 7N neuron (lower) showing spike rate increases in response to the DEC triggers (marked with CS and US) as well as spontaneous eyelid movements (red arrowheads). Right: average relative firing rate of all putative 7N motor neurons ($n = 19$, mean ± s.e.m) during spontaneous blinking (peak-aligned, red arrowhead). **c–e** Changes in behavior and 7N neuron activity in the control, IN-inhibition and FN-inhibition trials. Firing rate PSTH of an example 7N neuron (middle row) during behavior (upper row, $n = 46, 22, 26$ trials for **c–e**), showing that both IN and FN photoperturbations (blue bars) inhibited CR and neuron activity in response to CS, while only FN-inhibition suppressed UR and neuron activity in response to US. Lower: average spike modulation of all 7N neurons ($n = 19$, mean ± s.e.m). Dashed-line trace indicates the average activity during control trials in (**c**). **f** Summary of changes in relative firing rates of 7N motor neurons in response to CS and US in control, IN-opto and FN-opto trials ($n = 19$, mean ± s.e.m., paired two-sided $t$ test, **$P < 0.01$, ***$P < 0.001$). CR and UR modulation (ΔFiring rate) are normalized to the corresponding firing rate changes of the control session. See the exact $P$ values for each comparison in the Source Data file.

the facial nucleus, facilitating cooperation with the IN-RN pathway in regulating 7N motor neuron activity. Taken together, our findings indicate that the vermis-FN-MdV pathway plays a role in modulating both CRs and URs, while the well-established simplex-IN-RN pathway is the main circuitry driving CRs. These data highlight that conditioned and unconditioned sensorimotor behaviors can be controlled by different cerebellar modules in a distributed, yet synergistic manner.

**The vermis-FN-MdV module is essential for eyelid closure during DEC.** We found that excitatory FN neurons and vermal PCs had task-related modulation in response to a CS and a US, which is consistent with recent in-vivo calcium imaging studies, revealing the involvement of vermal (lobule V and VI) PCs and granule cells during DEC[44,45]. A subpopulation of these FN neurons and PCs might be recruited specifically for modulating the amplitudes of conditioned eyelid closure, as is evident from the trial-by-trial correlation between their activities and CR

amplitudes (Figs. 1, 3). In addition, we observed that CS-related modulations of FN neurons and vermal PCs were stronger in CR trials compared to those in non-CR trials, further supporting the task specificity of these neuronal activities in the vermal-FN module. Our data cannot completely rule out the possibility that some FN neurons and/or vermal PCs may encode other concurrent behaviors during DEC, including related body movements, preparatory muscle tone or vestibular signals. However, these behaviors possibly need to be controlled by the same group of cerebellar neurons, suggesting a synergistic coordination of different movements during DEC. Such concerted actions are in line with recent results from Heiney and colleagues who showed that neurons from the classic simplex-IN module also contribute to coordinating other body movements during DEC[60].

By using reversible pharmacological, optogenetic and chemogenetic interventions in FN, we show that the vermis-FN module is essential for both the acquisition and expression of CRs. These results are in line with a recent study from Giovannucci and colleagues (Supplementary Fig. 5 of ref. 45), showing that

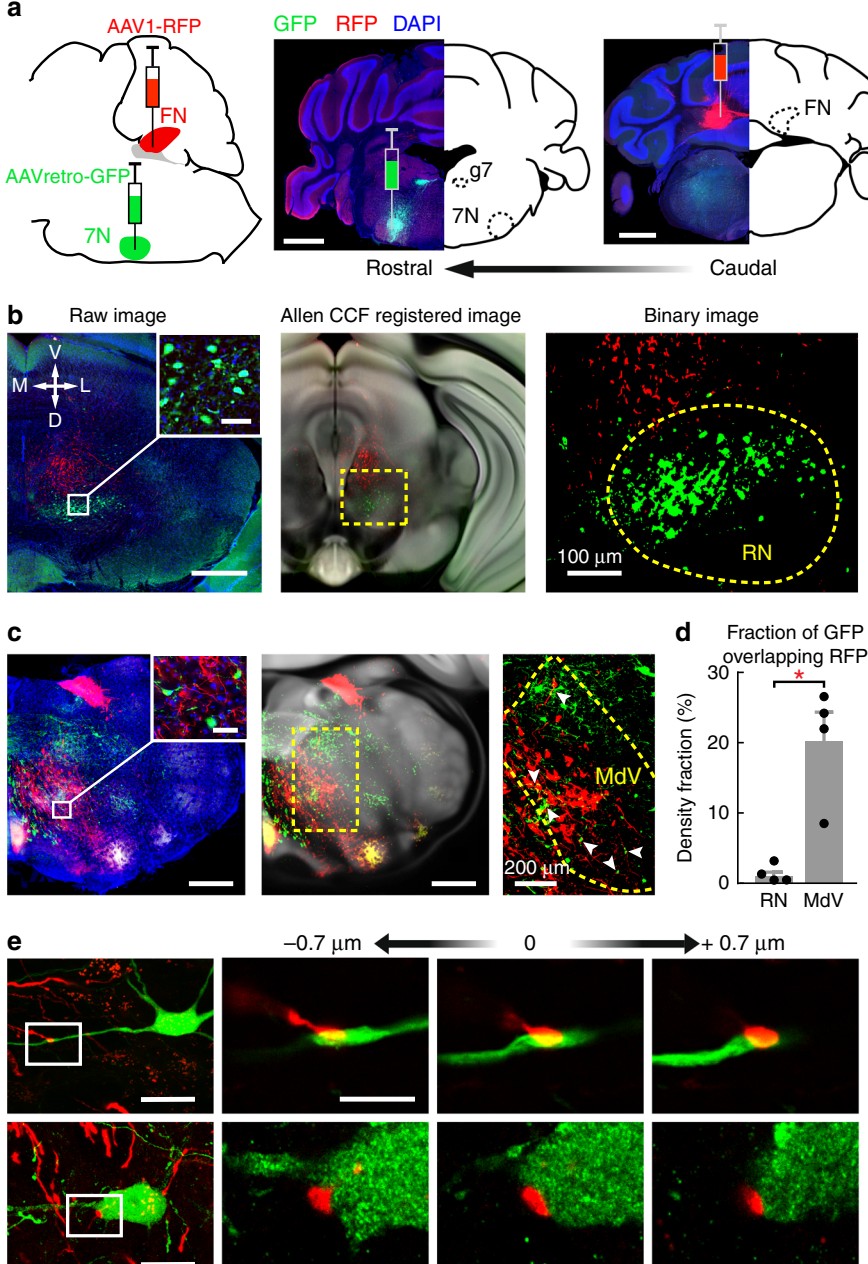

**Fig. 8 Anatomical tracing reveals a FN-MdV pathway for DEC. a** Sketch of the viral tracing strategy (left). Middle and right images: coronal sections of the injection sites from an example animal. Retrograde GFP and anterograde RFP were simultaneously injected into the ipsilateral facial nucleus (7N, middle) and FN (right). Labeled fibers in the genu of facial nucleus (g7) confirm the targeting of the 7N. Scale bars, 1 mm. **b** Representative images of retrogradely labeled neurons (green) and anterogradely labeled FN axons (red) at the level of caudal midbrain. The raw image (left) is registered to the Allen Mouse Brain CCF (middle, see "Methods") for further quantification of the FN projection in the contralateral red nucleus (right image, the RN is denoted by a dashed-line contour). Scale bars, 500 μm, inserted image 50 μm. **c** Same as (**b**), but for labeling in caudal medullary regions. Colocalizations of 7N projecting neurons and FN axons are found in the contralateral ventral medullary reticular nucleus (MdV, arrowheads in the right image). Scale bars, 500 μm, inserted image 50 μm. **d** Comparison of colocalizations in the RN and MdV from 4 mice (mean ± s.e.m., paired two-sided *t* test, *P = 0.020). **e** Confocal images of two example MdV neurons with FN axons targeting their primary dendrite (top) and soma (bottom). Scale bars, left column, 20 μm, right columns,10 μm. Tracing experiments were performed and replicated in *n* = 4 mice.

muscimol inhibition of vermal lobule VI (likely the area projecting to the FN) impairs CR amplitudes in trained mice[45]. The learning deficits were evident on the test day with FN inhibition omitted (Fig. 6f, i), suggesting that FN inhibition directly affects the associative learning process rather than merely deregulating eyelid muscle tone. Interestingly, we show that chronic FN lesions ipsilateral to the trained eye resulted in a significant and long-lasting impairment in CR performance.

Therefore, our study unequivocally highlighted the enduring relevance of FN output in sensorimotor tasks like DEC. Previous rabbit studies of chronic lesions in the FN and/or vermis have suggested that their DEC (nictitating membrane conditioning in rabbits) does not critically depend on an intact FN or vermal cortex[12,13]. This may be attributed to differences in the level of compensation after reversible perturbations and irreversible lesions, in the completeness of lesions, and/or in the kinematic

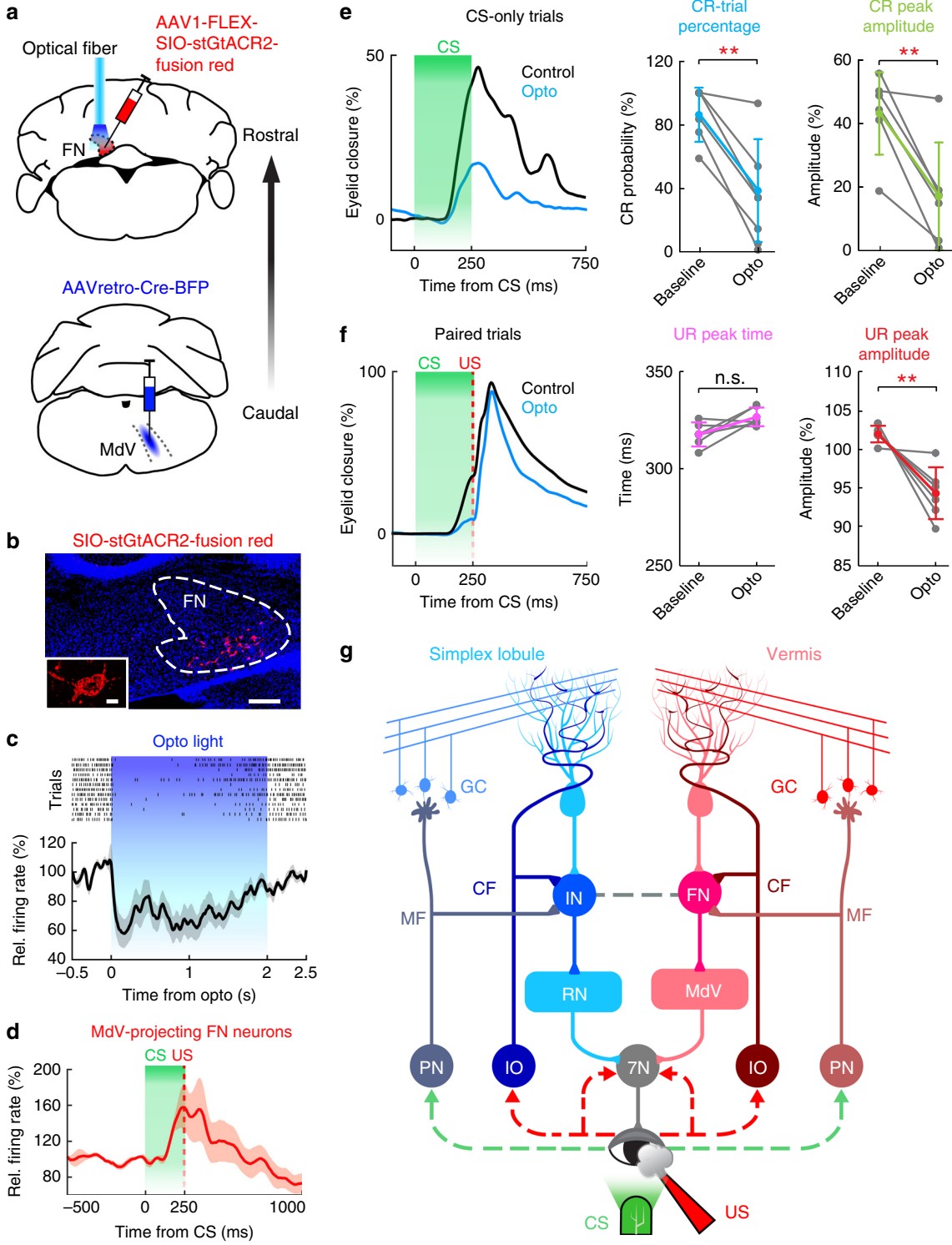

mechanisms of the conditioned eyelids in mice and the trained nictitating membrane responses in rabbits.

Furthermore, the anatomical elucidation of the vermis-FN-MdV pathway agrees with previous rabies tracing studies, revealing the cerebellar and brainstem regions that control eyelid movement by connecting the motor neurons of the *orbicular oculi* muscle[42,43]. Interestingly, in these rabies tracing studies, the MdV and RN appeared together as first-order labeled regions, ascending to facial motor neurons, whereas the FN and IN were found with coincidently labeled second-order connections,

indicating two parallel pathways that project to the motor neurons of the *orbicular oculi* muscle. Accordingly, the excitatory projection from the FN to the medullary reticular formation, specifically the contralateral MdV, may enable important functions in motor control, as the MdV has been proven to mediate skilled motor behaviors[61] and sensorimotor behaviors by modulating muscle tone[61–64]. In our experiments, targeted inhibition of the FN-MdV pathway impaired eyelid closure of both CRs and URs, suggesting that the vermis-FN-MdV pathway may be engaged during associative behaviors to modulate motor

**Fig. 9 Optogenetic suppression of the FN-MdV pathway affects CR and UR performance. a** Schematics showing viral injections and the optogenetic inhibition strategy to selectively identify MdV-projecting FN neurons and suppress the FN-MdV pathway during DEC. **b** Example image showing the somatic-targeting inhibitory opsin, stGtACR2, exclusively in FN neurons (insert). Scale bars, 200 μm, inserted image 10 μm, $n = 6$ mice. **c** FN neuron activity during optogenetic stimulation. Top: raster plot of an example neuron showing spike inhibition in response to optic light ($n = 13$ trials); bottom: average firing rate of all FN neurons in response to optic light ($n = 15$ neurons, mean ± s.e.m.). **d** Activity of the same photo-identified MdV-projecting FN neurons ($n = 15$ neurons, mean ± s.e.m.) in (**c**) during DEC. **e, f** Effects of optogenetic suppression of the FN-MdV pathway on behavior. Left column: average behavioral traces of an example mouse during CS-only trials (**e**) and CS–US paired trials (**f**). Middle and right columns: summary of behavioral performance. CR and UR amplitudes are significantly suppressed in trials with optogenetic perturbation ($n = 6$ mice, mean ± SD, paired two-sided $t$ test, **$P < 0.01$), while UR timing is not influenced compared to control trials ($n = 6$ mice, mean ± SD, paired two-sided $t$ test, $P > 0.05$). **g** Schematic drawing showing the proposed organization of the interposed nucleus (IN) and the fastigial nucleus (FN) pathways for DEC. Solid lines denote direct projections, and dashed lines indicate indirect innervations. Abbreviations: 7N, facial nuclei; CF, climbing fiber; GC, granule cell; IO, inferior olive; MdV, ventral medullary reticular nucleus; MF, mossy fiber; PN, pontine nuclei; RN, red nuclei. See the exact $P$ values for each comparison in the Source Data file.

commands sent to specific muscle groups. Here, we hypothesize that FN output is crucial for gating/modulating task-related movements that have to be acquired via a cerebellar learning process, yet the motor signal that directly drives conditioned eyelid closures is conveyed from the IN. This view is supported by our observation that various FN lesions and manipulations affect CR performance in trained mice. Yet, electric activation of the IN rather than FN neurons drives eyelid closure in naïve mice (Supplementary Fig. 10). Hence, although both the FN-MdV and IN-RN pathways project to 7N motor neurons, it is likely that their involvement in generating motor commands is fundamentally distinct.

**Shared and distinct neuronal dynamics in different cerebellar modules.** Decades of landmark studies on DEC have achieved an unprecedented understanding of the cerebellar circuits for associative learning and behavior[3,5–8,10–14,17,19,22–24,31,32,52,54–56,65]. PCs from the canonical simplex lobule-IN module have been shown to primarily modulate CRs by suppressing their simple spike activity and increasing their CS-related complex spike activity[10,19,28]. Our recordings from vermal PCs reveal very similar activity patterns, showing that simple spike suppression correlated well with the conditioned eyelid amplitude. In addition, these vermal PCs showed CS-related complex spikes that correlated well with the onset timing and amplitude of the CRs, similar to those in the simplex lobule[17,22]. We observed PCs in both the simplex lobule[19] and the vermis that show simple spike facilitation in response to a CS. The simple spike facilitation of these PCs had a weaker correlation with the CR amplitude than that of simple spike-suppression cells. At present, it is unclear what information simple spike facilitation might encode during eyeblink conditioning. In principle, they may control antagonistic eyelid muscle relaxation or synchronize other movements that occur concomitantly during the CS–US interval.

In line with the notable role of vermal PCs in DEC, we showed that facilitating FN neurons correlates well with the timing and amplitudes of CRs, which is reminiscent of how IN neurons probably control CRs[8,17]. However, unlike IN neurons showing virtually exclusive positive correlations[17], the activity of about half of the facilitation FN neurons showed a negative trial-by-trial correlation with CR amplitudes. Therefore, it is conceivable that the vermal PCs and FN neurons may comprise antagonizing functional groups, together actively regulating both the closure and the opening of eyelids. These differences in activity dynamics may stem from the specific input information to these cerebellar modules. Our retrograde tracing in the simplex lobule and vermis showed distinctly labeled, adjacent regions in the inferior olive and pontine nuclei (Supplementary Fig. 11f–h), suggesting that distinct climbing fiber and mossy fiber inputs to the cerebellar modules may contribute differently to associative behaviors.

Despite the occurrence of the opposite correlations highlighted above, together with our previous work[17,19], most of the task-related FN and IN neurons showed increased activity during DEC[19]. This finding is in line with the fact that CR amplitudes tend to correlate best with simple spike suppression in both the vermal and the simplex PCs[19]. In this regard, it is intriguing that the proportion of FN neurons showing facilitation during DEC (approximately 53%) mismatches that of the vermal PCs with CS-related simple spike suppression (approximately 37%). Using multichannel silicon probes, we sampled FN neurons and vermal PCs in an unbiased manner. Due to the high PC-cerebellar nucleus convergence ratio[66], the chance is high that more task-irrelevant PCs were recorded. Second, sensorimotor information conveyed by the excitatory mossy fiber and climbing fiber collaterals[22,54,67] may also directly facilitate FN neurons during DEC; hence, it is possible that inputs from specific vermal PCs and/or precerebellar mossy fiber and climbing fiber sources contribute to the relatively dominant facilitation of FN neurons during DEC. Further study is required to clarify the roles of mossy fiber and climbing fiber collaterals in the vermis-FN module during associative learning and behavior.

The shared neural dynamics between the vermis-FN module and simplex lobule-IN module suggest that common inputs might facilitate synergy across different functional modules to a certain extent. With regard to the mossy fiber inputs, our results show that there is minimal overlap in their resources in the pontine nuclei. However, the mossy fibers innervate the granule cells that give rise to parallel fibers traversing many zones, which may well reach beyond the cerebellar vermis and hemispheres[68,69]. A common relay of CS signal by beams of parallel fibers may be further corroborated via cerebellar nucleo-cortical feedback loops to the granule cell layer[23,70,71], which indeed has been implicated to generate representations of predictive signals during DEC[45]. Likewise, the climbing fiber sources innervating the eyeblink regions in the vermis and hemispheres do not overlap in the inferior olive. Yet, here too, one must be aware of the fact that DAO neurons projecting to the vermal eyeblink region are located just caudal to the neurons in the DAO that provides climbing fiber inputs to the simplex eyeblink region[72,73]. It is exactly this transition zone between the DAO and principal olive that receives prominent inputs from the trigeminal nucleus, i.e., the main source mediating US signals during DEC[9,19,22,31,55]. In addition, given the proximity of these two regions, these IO neurons are likely to be electrotonically coupled by dendrodendritic gap junctions[72], further facilitating convergent multifunctional signaling[74]. In conjunction, this configuration may well explain why we found very similar short-latency complex spike responses in the vermis and lobule simplex following a US[72]. Thus, both the simplex lobule and vermis may well have access to very similar CS and US signals.

**Multimodular control of sensorimotor tasks and functional implications**. Acquisition and expression of DEC require sophisticated and well-timed cerebellar coordination of attention, preparatory muscle tension, autonomic responses and concurrent body movements that systematically accompany eyelid movements[10,60,75]. A previous study has shown that associative learning induces the formation of new synapses in both the IN and the FN, indicating the structural plasticity of mossy fiber inputs in multiple cerebellar modules[75]. Current data highlight the possibility that the activity of different cerebellar modules can modulate behavior simultaneously during an associative sensorimotor task. Activity from both the simplex-IN module and the vermis-FN module is a prerequisite for generating conditioned eyeblink responses, suggesting that neither pathway is functionally redundant. Notably, the UR amplitude can be significantly affected by inhibiting the FN pathway rather than the IN pathway. These results imply that inhibiting vermis-FN-MdV activity generally deregulates the output of facial motor neurons. As FN neuron activity is prominent during DEC-related eyelid closure, but not during spontaneous blinking, this modulation appears to play a role specifically for acquired sensorimotor behaviors. Taken together, we hypothesize that CS-related activity of the simplex lobule-IN-RN pathway serves exclusively as the driver for initiating CRs, while acquired task-related modulation of the vermis-FN-MdV pathway may gate or fine-tune the excitability of facial motor neurons, which is critical for both conditioned and unconditioned reflexes.

In addition, task-related FN output could impose additional control over the IN pathway during DEC (Fig. 5j–l). The FN projects to a myriad of downstream targets that play various roles in both motor and nonmotor functions[36–41]. In this study, we mainly focused on the FN-MdV pathway, yet it is possible that MdV-projecting FN neurons provide efference copies to many other brain regions by their collateral projections[76]. We observed that inhibiting FN output could directly affect task-related IN activity. This cross-modular effect indicates the presence of circuits for the synergistic control of different cerebellar modules, possibly via cerebellar nucleo-cortical feedback loops[23,70,71], cerebro-cerebellar loops[38,77,78] or other brain regions that are currently unknown. Future examination of functional synergy across different cerebellar modules appears crucial to fully comprehend cerebellar coordination of sensorimotor behaviors.

## Methods

**Mice**. All animal experiments were approved by the institutional animal welfare committee of Erasmus MC in accordance with the Central Authority for Scientific Procedures on Animals guidelines. Wild-type C57BL/6J (No. 000664) and transgenic Gad2-ires-Cre (No. 010802), VGluT2-ires-Cre (No. 016963), L7-Cre (No. 004146) and Ai27D (No. 012567) mice were obtained from the Jackson Laboratory. All mice in this study were 6–14 weeks old and were housed individually in a 12-hours light-dark cycle with food and water ad libitum. Ambient housing temperature was maintained at ~25.5 °C with 40–60% humidity. We used 37 mice for fastigial and Purkinje cell recordings for Figs. 1–4, and each mouse contributed to multiple datasets except where additionally indicated. We used 68 mice for behavioral, tracing and pathway specific perturbation experiments in Figs. 5–9.

**Viral vectors**. Adeno-associated virus AAV9-Syn-FLEX-ChrimsonR-tdTomato, AAV2-hSyn-DIO-hm4D-mcherry, AAV9-hSyn-FLEX-tdTomato and AAV5-hSyn-Cre-eGFP were obtained from UNC Vector Core. AAVrg-CAG-GFP, AAV1-CB7-CI-TurboRFP, AAV1-CB7-CI-eGFP, AAVrg-Cre-eBFP and AAV1-hSyn1-SIO-stGtACR2-FusionRed were obtained from Addgene. All viral vectors were aliquoted and stored at −80 °C until used.

**Surgical procedures**. Mice were anesthetized with 5% isoflurane for induction and 2.5% for maintenance. Animals were fixed on a mouse stereotaxic surgical plate (David Kopf Instruments) with eyes covered with DuraTears (Alcon Laboratories, Inc.), and body temperature was kept at 37 ± 0.5 °C constantly during operation. We injected bupivacaine (4 mg/kg) intraperitoneally after surgery.

For pedestal placement and craniotomy operation in all the behavioral or/and electrophysiology experiments, after removing hair over the scalp, we sprayed

lidocaine (2.5 mg/mL) locally on the skin, and a vertical skin cut was applied to expose the skull. The skull was pretreated with Optibond All-in-one (Kerr), and a 5.5 × 4.0 mm custom-made pedestal was attached to the skull with Charisma (Heraeus Kulzer). Mice were allowed to recover for at least two weeks after surgery and prior to behavior studies. For in-vivo electrophysiological experiments, a small craniotomy (Φ = 2.0 mm) was made on the skull over the recording sites. We built a chamber around the craniotomy with Charisma and sealed it with Picodent twinsil after recording.

For intracranial viral/CTB injections, anesthetized animals were fixed on a mouse stereotaxic surgical plate. The skull was exposed, and the head was positioned so that the bregma and lambda were leveled. See the coordinates for different brain regions in Table S1. We gently lowered the glass capillary (tip opening Φ = 8 µm), and AAV viral vector/CTB was slowly injected in the targeted regions. Glass capillaries were left on the injection sites for > 5 mins before slowly retracted from the brain. To optogenetically perturb FN and/or IN neurons, a 2 mm long optical fiber (Φ = 200 µm, 0.22 NA, ThorLabs) was inserted through a small craniotomy (Φ = 300 µm) and was chronically fixed to the skull with Charisma.

**Behavioral training**. Mice were head-fixed and suspended on a cylindrical treadmill in a light- and sound-attenuated chamber. A light-emitting diode was presented approximately 7 cm in front of the animal as the conditioned stimulus (CS), together with a corneal air puff (tip opening Φ = 1.5 mm, 30 psi) placed approximately 1 cm to the left eye as the unconditioned stimulus (US). A paired trial is consisted of a 250 ms CS light, coterminating with a 10 ms US air puff. Mice were trained with 200 paired trials with 10–15 s randomized inter-trial intervals for 7–10 days. In some cases, CS-only and US-only trials were delivered every 10th trial. Eyelid position was real-time monitored by a 250 fps camera (scA640-120gc, Basler) and was digitized and acquired through a RHD2000 Evaluation System (Intan Technology) at 20 kHz sampling rate. Triggers for CS and US were controlled by a NI-PXI system (National Instruments) with using custom-written Labview codes.

**In-vivo electrophysiology**. We used single-channel or multichannel electrophysiological acquisition system in this study. We recorded vermal Purkinje cells at a depth of 1.5–2.0 mm and FN neurons at a depth of 2.0–2.7 mm, as measured from the cerebellar surface. For single-channel recordings, a glass capillary (tip opening Φ = 2 µm) filled with 2 M NaCl solution was slowly penetrated in the cerebellum till a well-isolated neuronal signal was observed. Neuronal signals were notch-filtered at 50 Hz, amplified and digitized at 20 kHz sampling rate by using Axon acquisition system (1440 A, Molecular Devices Corporation). Multichannel recordings (32-channles ASSY-32-E2 or 64-channles ASSY 77H-H2, Cambridge NeuroTech) were amplified and digitized on an Intan RHD2000 Evaluation System (Intan Technology) at 20 kHz sampling rate and were further analyzed offline using custom-written Matlab codes. In all electrophysiological experiments combined with DEC, at least 50 CS–US paired trials were given to animals, and neurons with a minimum of 20 CR trials were included in the datasets. In the recordings with optogenetic perturbations, at least 15 optogenetic trials were delivered.

**Optogenetic manipulation**. Electrophysiological recordings were carried out four weeks after AAV injection and/or optical fiber implantation. In all the optogenetic experiments, manipulations were applied in the FN/IN ipsilateral to the trained eye. We used an orange LED light source (M595F2, Thorlabs) to activate ChrimsonR and a blue LED light source (M470F3, Thorlabs) to activate ChR2 and SIO-stGtACR2. Optical light with 100 Hz pulse, 50/50 duty cycle was controlled by a high-power light driver (DC2100, Thorlabs). Optical fiber was wrapped with light-isolating aluminum foil so that mice would not perceive the optogenetic light as a CS.

To express ChrimsonR specifically in excitatory or inhibitory FN neurons, AAV9-Syn-FLEX-ChrimsonR-tdTomato was injected in the FN of VGluT2-Cre or Gad2-Cre mice. To identify the ChrimsonR-expressing neurons in-vivo, we first illuminated the orange light (125 ms, 4.5 mW) and recorded the neuronal responses in the FN. Only neurons with short-latency responses to the optogenetic stimulation (latency < 20 ms)[79,80] were included in the datasets for further analysis of their responses during behavior.

We used L7Cre-Ai27 mice to study the effects of activating Purkinje cells, hence, suppressing FN or IN neurons, on the acquisition and expression of DEC. Light intensity was adjusted to 1–1.5 mW so that no obvious aversive behavior, locomotion impairment or silencing of neighboring IN neurons were observed, similar to what was reported in our previous work[38]. In well-trained animals (Fig. 5d–l), at least 20 optogenetic trials, consisting of 250 ms blue light flanking the CS and US epochs, were randomly presented in 50 control trials (no photoperturbation). To examine the effects of optogenetic perturbation during learning (Fig. 6g–i), naive animals were daily trained for 200 trials with the same optogenetic condition mentioned above, for 10 consecutive days. On day 11, we omitted the optogenetic perturbation to test the effects of optogenetic perturbation on learning. In order to suppress FN or IN output while recording 7N neurons (Fig. 7), two optic fibers were implanted into the FN and IN of L7Cre-Ai27 mice. During recording, at least 20 FN-opto trials and 20 IN-opto trials were randomly

placed in 40 control trials (no photoperturbation), and we targeted putative 7N motor neurons based on the stereotaxic coordinates (Supplementary Table 1) and their stereotypical firing patterns in response to eyelid movements.

To specifically suppress the FN-MdV pathway in well-trained animals (Fig. 9), an inhibitory opsin stGtACR2 was expressed exclusively in the MdV-projecting FN neurons by simultaneously injecting a Cre-dependent AAV1-hSyn1-SIO-stGtACR2-FusionRed in the FN and a retrograde AAVrg-Cre-eBFP in the MdV. At least 15 optogenetic trials with blue light (4.5 mW) flanking the CS and US epochs were randomly placed in 50 normal CS–US paired trials (control) to test behavioral changes following FN-MdV pathway inhibition. We identified the stGtACR-expressing FN neurons by in-vivo recording the spike rate changes in response to optogenetic illumination, and further tested the activities of these "opto-tagged" FN neuron during DEC.

**Chronic photolesions of FN.** Nine animals were trained with 200 DEC trials daily for 10 days. We randomly divided these conditioned mice into two groups ($n = 4$ for FN lesion and $n = 5$ for sham surgery). CR performance on pre-lesion day 11 had no significant difference between two groups (two-sample two-sided $t$ test, $P > 0.05$). For photolesioning FN (ipsilateral to the conditioned eye), an optical fiber (200 μm diameter, 0.22 NA, Thorlabs) was inserted into the FN. Continuous blue light (15–30 mW) was applied for 10 min to lesion the FN. The control group (sham group) underwent the same surgical procedures without laser application. After photolesioning, we tested the CR performance for three consecutive days. Animals were sacrificed for histology to confirm their lesion sites at the end of the experiment.

**Electrical activation of FN/IN regions.** Craniotomy were made above the FN/IN stimulation sites. A stimulation glass electrode (tip opening = 8 μm) filled with 2 M saline-0.5% alcian blue was lowed into the FN or IN based on their stereotaxic coordinates (Supplementary Table 1). Animals received stimulation (250 μs biphasic pulses, 200 ms pulse train, 500 Hz) in either the FN or IN sequentially starting from 0.2, 0.6, 0.8, 1.0 μA to maximum 1.2 μA. Eyelid movement was recorded and digitized as described above. Mice were sacrificed for histology after stimulation to confirm the stimulation locations.

**Pharmacological and chemogenetic Inhibition.** All the pharmacological and chemogenetic inhibition experiments were performed in the FN ipsilateral to the conditioned eye. To inhibit FN neurons in trained mice during DEC, we performed a craniotomy on FN three days before behavioral tests. A glass capillary with 0.05% muscimol (Tocris Bioscience, 0289)-0.5% alcian blue (volume ratio 1:1) was lowered into the FN region, and about 10 nL mixture was injected 5 min before DEC tests. Animals were sacrificed for histological check of the injection site. We expressed inhibitory DREADDs in FN neurons by coinjecting AAV2-hSyn-DIO-hm4D-mcherry and AAV5-hSyn-Cre-eGFP (volume ratio 1:1) in the FN. CNO (Santa Cruz, sc-391002A) was dissolved in 2.5% dimethyl sulfoxide (DMSO) as stock (light shielded at 4 °C) and diluted with saline as working solution (0.6 mg/mL). Animals were intraperitoneally administered with CNO working solution (3 mg/kg) 15–20 min before training sessions for 10 days. On day 11, we omitted the CNO administration to test the learning outcomes. In order to prove effectiveness of the DREADD-CNO system, we recorded FN neuron activity changes over time after the CNO injection in awake mice (Fig. 6b, c).

**Histology and microscopy.** For vermal Purkinje cell retrograde tracing, goat ati-cholera toxin B subunit primary antibody (1:15000, List labs, 703) and biotinylated horse anti-goat secondary antibody (1:2000, Vector, BA-9500) were used. For VGluT2 staining, guinea pig anti-VGluT2 primary antibody (1:750, Sigma-Aldrich, AB2251-I) and Alexa fluor® 488 donkey anti-guinea pig secondary antibody (1:400, Jackson, 706-545-148) were used. For Gad2 staining, rabbit anti-Gad65/67 primary antibody (1:1000, Sigma-Aldrich, AB1511), and Alexa fluor® 488 donkey anti-rabbit secondary antibody (1:400, Jackson, 711-545-152) were used. For GFP staining, goat anti-GFP primary antibody (1:5000, Rockland, 600-101-215) and Alexa fluor® 488 donkey anti-goat secondary antibody (1:200, Jackson, 705-545-147) were used. For NeuN staining, rabbit anti-NeuN primary antibody (1:1000, Millipore, abn78) and Cy™3 donkey anti-rabbit secondary antibody (1:400, Jackson, 711-165-152) were used.

Animals were deeply anesthetized with intraperitoneal injection of pentobarbital sodium solution (50 mg/kg) and perfused transcardially with saline, followed by 4% paraformaldehyde (PFA) in 0.1 M phosphate buffer (PB, pH 7.4). Brains were removed immediately and post-fixed overnight in 4% PFA-0.1 M PB at 4 °C. Fixed brains were placed in 10% sucrose overnight at 4 °C and embedded in 12% gelatin-10% sucrose. After fixation in 10% formalin-30% sucrose overnight at 4 °C, serial coronal sections were cut with microtome (SM2000R, Leica) at 40 μm and collected in 0.1 M PB. For immunohistochemistry and immunofluorescence purposes, sections were incubated subsequently with primary and secondary antibodies (see titrations above). All antibodies were titrated for working solution with 2% normal horse serum-0.4% triton-0.1 M PBS solution. Primary antibodies were incubated at 4 °C overnight, and secondary antibodies were incubated at room temperature for 2 h. After each incubation session, sections were gently rinsed with 0.1 M PBS (10 min, 3 times). For CTB immunohistochemistry, rinsed sections were additionally incubated in avidin-biotin complex (volume ratio 1:1, Vector, AK-5200) for 1.5 h at room temperature. Next, labeling was visualized with DAB staining (1:150), and sections were mounted for microscopy. For all immunofluorescence sections, DAPI was used for general background staining. Bright-field images were captured with Nanozoomer (2.0-RS, Hamamatsu). For fluorescence imaging, we took overviews of the brains with a 10x objective on a fluorescence scanner (Axio Imager 2, ZEISS) or high-magnification images on a confocal microscope (LSM 700, ZEISS). Images were post-processed with ImageJ and Adobe Photoshop.

**Behavioral analysis.** As described in our previous work[17,23], to investigate the eyelid position changes in response to the CS and US, each trial was normalized to a 500 ms baseline prior to the CS onset. We removed trials with noisy baseline (spontaneous blinking) by performing an iterative Grubbs' outlier detection test ($\alpha = 0.05$) on the standard deviations of baseline. A CR trial was determined if eyelid closure exceeded 5% of the mean baseline, and CR onset was defined as the timing which eyelid closure exceeded three SDs of the baseline value. Peak amplitudes of CR and UR were detected in a 50–250 ms window during the CS–US interval and a 100 ms window after US, respectively. To estimate the 7N neuron activity changes in response to spontaneous blinking, we detected spontaneous blinking events with prominent peak heights (> 10% maximum blink peak of the recording session) and aligned them at peak epoch. The sample size of spontaneous blink in each recording session was at least 20, which statistically meets the requirement for analyzing the corresponding neuron modulation.

**Electrophysiological analysis.** Single-channel recordings and Purkinje cell complex spikes were analyzed by using in-house developed code SpikeTrain (Neurasmus) in Matlab[17,19]. Raw recordings were band-pass filtered at 300–3000 Hz to subtract noise and field potential signals. We extracted spike events with amplitudes that crossed the threshold at three SDs of the baseline noise. We performed additional manual waveform sorting to Purkinje cell complex spikes based on the distinct features of an initial spike followed by high-frequency spikelets. Neurons from multichannel recordings were sorted with JRCLUST[81], and all spike time was stored for further analysis. In order to determine the cell modulation is response to CS, only cells with more than 20 CR trials were included in the dataset. Peristimulus time histograms (PSTHs) of well-isolated units were constructed by superimposing the CS onset-aligned spike time in a shifting window (50 ms window size, 5 ms increment) which shifted from 500 ms before CS and onward, and were expressed as frequency. We conservatively performed CR modulation detection within 50–200 ms after CS onset due to the fact that the shifting window for PSTH construction went across CR and UR in the last 50 ms of CS. Baseline firing rate was calculated as the mean frequency in a 500 ms window prior to the CS onset. We determined firing rate changes in response to CS by subtracting the baseline firing rate from the frequency within 50–200 ms after CS onset. Cells with average firing rate changes (50–200 ms) more than three SDs of the baseline frequency were considered modulating in response to CS. We then discriminated the direction of modulation by linearly fitting the PSTH of spike frequency within the 50–200 ms window. Fittings with positive slopes were considered as facilitation, whereas with negative slopes were considered as suppression. UR modulation was determined within the first 100 ms after US. As previously described[17], spike rate change more than 5 Hz after US was considered as US-related modulation,.

**Correlations between neuronal activity and behavior.** To compare the temporal timing of neuron activity and behavior, PSTH of firing frequency (5 ms bin width) were generated without a shifting window. CR modulation onset was defined as the corresponding timing of the first value of PSTH (50–250 ms after CS onset) at which was more than three SDs of the baseline frequency. Modulation peaks were defined as the maximum values within the 50–250 ms window (CS-related) and the 250–350 ms window (US-related) after CS onset. To investigate the relationship between CR peak amplitudes and neuron activity on a trial-by-trial basis, we calculated the instantaneous firing frequency by summing spike counts in a 50 ms shifting window with a 5 ms increment. Across trials, CR peak amplitudes and extremums of firing rates (maximum for facilitation, minimum for suppression) were correlated by performing a linear regression model. To construct the temporal correlation matrix for multiple cells, we calculated the average $r^2$ values of eyelid closure-neuron activity correlations in a −250–500 ms window (CS onset as 0 ms) and illustrated in heat maps. The most-correlated pixels were accentuated by comparing with all $r^2$ values in the matrix neglecting their significance.

**Anatomy analysis.** To quantify the fluorescence signals from our tracing experiments, we first registered slices into the Allen Mouse Common Coordinate Framework (CCF)[82] in order to standardize the brain slices across mice and to annotate nuclei. Detailed registration method has been described in our previous work[38]. Briefly, we manually selected the coronal plane from the CCF template (10 μm per voxel) that best corresponded to our section. Next, at least 30 control points were placed at the corresponding locations of section and CCF template. Sections were warped to the CCF template by using an affine transformation followed by a non-rigid transformation b-splines[83]. We quantified the connectivity of FN axons (RFP) and retrogradely labeled cells (GFP) in the RN and the MdV (Fig. 8b–d) by

transforming the registered images into binary images. We then thresholded the fluorescence intensities for both channels at 85th-95th percentiles. Fractions of GFP overlapping RFP in the RN and the MdV were calculated with ImageJ and presented in individual mouse for statistics. To quantify the projection densities from the FN and IN in the MdV and RN (Supplementary Fig. 11a–e), same threshold (85th-95th percentiles) was applied independently to both channels of binary images, and area fractions were calculated for both FN and IN projections and presented in individual mouse for statistics. To quantify the common inputs projecting into the simplex lobule and the vermis (Supplementary Fig. 11g, h), we manually counted the retrogradely labeled cells in the inferior olive and the pontine nuclei and presented the data in individual animal for statistics.

**Statistics**. All statistics were performed by using Matlab, SPSS and GraphPad Prism. Behavior was illustrated as average of all trials ± standard deviation (SD). The neuron frequency changes were plotted as the average of all cells ± standard error of mean (s.e.m.), and sample sizes were displayed in the figure legends. Statistical comparisons were performed by using $t$ tests, repeated measures ANOVA and restricted maximum likelihood model depending on the experiment and data specificity, unless stated otherwise. Statistical significance was defined as $P < 0.05$, and annotations were $*P < 0.05$, $**P < 0.01$, $***P < 0.001$ respectively. No significant difference was denoted as n.s.

**Reporting summary**. Further information on research design is available in the Nature Research Reporting Summary linked to this article.

## Data availability

Source data are provided with this paper. Please see data for all figures in the Resource Data file. The raw datasets generated and analyzed in the current study are available from the corresponding author (Z. Gao) upon reasonable request. Source data are provided with this paper.

## Code availability

All custom analysis codes generated in Matlab can be found in the following repository: https://github.com/XiaoluOne/FN-eyeblink. The data acquisition codes created in Labview, and other custom codes in Matlab are available from the corresponding author (Z. Gao) upon reasonable request.

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

## Acknowledgements

The authors thank M. Rutteman (Department of Neuroscience, Erasmus MC) for taking care of the animal breeding; H. Boele and S. Dijkhuizen (Department of Neuroscience, Erasmus MC) for providing assistance for behavioral training; G. Borst and S. Kushner (Department of Neuroscience, Erasmus MC) for sharing virus for the DREADD experiments; and N. Li (Department of Neuroscience, Baylor College of Medicine), D. Jaarsma, H. Hasanbegovic, and C. Schafer (Department of Neuroscience, Erasmus MC) for suggestions on the manuscript. This work was supported by CSC fellowship (Z.R.); ERC advanced, ZonMw, NWO-ALW, Medical NeuroDelta, INTENSE (LSH-NWO), Trustfonds, van Raamsdonk fonds, Vriendenfonds Albinisme, and BIG (C.I.D.Z.); EUR fellowship, Erasmus MC fellowship, NWO VIDI, NWO-Klein and ERC-stg grants (Z.G.).

## Author contributions

X.W. and Z.G. conceived the project and designed the experiments. X.W. and Z.R. performed all the experiments. X.W. and Z.G. conducted anatomical analysis. S.Y. and X. W. developed the algorithm for electrophysiological and behavioral analysis. X.W. analyzed the electrophysiological and behavioral data. X.W., C.I.D.Z., and Z.G. wrote the manuscript with inputs from all authors. C.I.D.Z. and Z.G. supervised the project.

## Competing interests

The authors declare no competing interests.
