## [Peer Review File · Nature Communications]

REVIEWER COMMENTS

Reviewer #1 (Remarks to the Author):

In general, the authors are to be commended for the additional experiments, clarification of technical details, and the more appropriate discussion regarding their conclusion that the FN circuit seems to play a modulatory role in delayed eyeblink conditioning.

In particular, the revision provides a lot of new data, and alternatives like control of posture/non-eyelid movements are mentioned in the discussion. Results of new experiment showing that FN electrical stimulation does not cause eyelid movement create some complications given the main hypothesis, leaving open the important question how these midline circuits appear to be essential. This is still confusing to me and I expect will be to many readers. In a revised version it would be helpful to spell out how this could work.

Reviewer #2 (Remarks to the Author):

My previous review when this paper was under consideration for [REDACTED] Neuroscience highlighted a few areas where the manuscript could be improved, namely:

- Doing a causal manipulation (electrical or opto stimulation) in FN to determine whether activation of these neurons can directly drive an eyeblink.
- Doing a chronic lesion in FN to see if the deficits in CRs and URs remain to rule out that the deficits were due to acute perturbations of the circuit dynamics.
- Toning down the language about the strength of the trial-by-trial correlations, given that the number of neurons with significant correlations is low.
- Addressing the (unexpected) result that inhibiting FN caused modulation of activity in IN.
- Discussing more fully the high level idea the authors are proposing that the FN "dynamically modulates muscle tone" during DEC.

While it would have been easier to evaluate how well the authors addressed these points (and the more minor ones) if they had tracked changes in the manuscript document or pointed to specific pages/line numbers in their rebuttal (I acknowledge that this may have been at least partially due to the transfer of the submission from Nat Neuro to Nat Comm), I am satisfied that the authors addressed them well enough to warrant publication in Nature Communications.

However, there are a few remaining points I think the authors should address before publication to improve the readability and completeness of the manuscript.

1. Figure 1: I may be missing some critical detail but I'm having trouble understanding why a portion of "facilitated" neurons is negatively correlated with CR amplitude (Fig 1h). Isn't the authors' definition of a "facilitated" neuron that it increases activity during the CR? Is it really the case that some neurons fire a bunch for really small CRs and then fire less strongly (but still above baseline) for bigger CRs? This would be pretty strange but interesting if true. I found this part very confusing. Please clarify.
2. Results related to Fig 3, page 8, second paragraph: Are the n's reported (8 and 1) only those neurons that were significantly correlated? Out of how many total tested? Please clarify.
3. How many of the suppressed PCs had short latency CpxUS? Did any of the facilitated PCs have short latency CpxUS? The authors mostly focus on the CS-related complex spike but the US-related complex spike is the canonical IO signal that has been used for defining eye blink-related cortex zones so I would like to see a little more treatment of this signal in the Results.

4. While I agree with the authors' point about the limitations of chronic lesions, I could just as easily apply a similar logic to disruptions of circuit dynamics during acute manipulations, so I don't find their rebuttal of this point totally compelling. Is there any other data they can provide to make the case that the particular part of the FN they have studied is uniquely contributing to DEC, rather than the alternative that "disrupting a major cerebellar output causes general circuit dysfunction"? I'm not proposing that they do additional experiments but just hoping that they have some existing data that could address the point. Perhaps from "misses", where they manipulated a different part of the cerebellum and didn't abolish CRs/URs? In my opinion, lack of these data should not prevent this manuscript, which was otherwise a very technically rigorous set of experiments, from being published, but I think it would at least warrant the authors addressing this caveat in the Discussion. And if they have the data it would greatly strengthen their conclusions.

Reviewer #3 (Remarks to the Author):

In the previous round of reviews, all three reviewers shared similar concerns about the functional relevance of the FN pathway for eyeblink conditioning. These concerns included:

- an extensive literature exists which argues that the IN pathway is sufficient and the FN is not needed for CR's. To address this, the authors have "analyzed the activity of FN neurons and vermal PCs in CR trials and non-CR-trials, as well as during spontaneous blinks (Supplementary Fig. 2, 5)." And argue that these results, "suggest that CS and US inputs to the vermis-FN module are not engaged during spontaneous blinks," (OK), "but are crucial for controlling eyelid closure during the task...". I don't see how these data speak to the FN being "crucial".

- the possibility that the vermal/FN signals are not specifically related to eyelid movements (Reviewers 1 and 3). Here the authors make some arguments about response latencies but ultimately admit that their data "at least raise the possibility" that these signals are related to eyelid movements.

- the use of only acute inhibition (muscimol, DREADDs) that could acutely disrupt activity in cerebellar target regions, and no chronic manipulations (Reviewer 2). The authors expressed concern that if they did these experiments, the results might be difficult to interpret. But given the valid concerns about the acute manipulations, we can only really judge that once the experiments are done and the data presented.

- the weak correlation between the kinematics of the eyelid closure and the neural signals (Reviewers 2 and 3). The authors "toned down the claim by using 'significant correlation' instead of 'strong/clear correlation'," and they argue in the rebuttal that the small percentages of modulated neurons are justified by an entirely unrelated recent study. They have not really responded to my concern about a lack of analysis or demonstration of trial-to-trial correlations of the timing of the responses and the behavior.

- the nature of the FN modulation of learning (all Reviewers). The authors object to the characterization of the FN role as 'permissive,' even though it was their exact language in the original manuscript. They have changed the language, but in my view these concerns, raised as central points by all three reviewers, remain.

Rebuttal letter

We highly appreciate the valuable and constructive comments of the reviewers. For the new version of the manuscript, we have performed the requested chronic FN lesion experiments and revised the main text and figures accordingly. We hope all reviewers now find our new data convincing and the manuscript acceptable for Nature Communications.

Reviewer #1:

In general, the authors are to be commended for the additional experiments, clarification of technical details, and the more appropriate discussion regarding their conclusion that the FN circuit seems to play a modulatory role in delayed eyeblink conditioning. In particular, the revision provides a lot of new data, and alternatives like control of posture/non-eyelid movements are mentioned in the discussion. Results of new experiment showing that FN electrical stimulation does not cause eyelid movement create some complications given the main hypothesis, leaving open the important question how these midline circuits appear to be essential. This is still confusing to me and I expect will be to many readers. In a revised version it would be helpful to spell out how this could work.

We thank the reviewer for her/his compliments and support. Our results indeed show that IN activation, but not FN activation, directly drives eyelid closure in naïve mice. However, acutely inhibiting IN activity or FN activity both impair CR performance (see Fig. 5), which is now even confirmed by the chronic lesion experiments (see Fig. 5j-o). In line with other perturbation results, both the CR-trial percentage and amplitude were significantly reduced after FN lesions. These impairments last for all three post-lesion testing days without noticeable recovery, leaving little question about the involvement of the FN pathway in DEC.

We interpret these findings as follows: Associative DEC requires coordination of multiple cerebellar modules to optimize the motor output in awake behaving mice. Our study is consistent with previous literature that IN provides the driver output for DEC, but suggest that acquired FN activity is also important for further modulating the learned eyelid closure, including the related muscle tone. Indeed, the differential roles of IN and FN are also reflected by the fact that inhibition of FN, but not IN, affects the amplitude of the unconditioned response. Thus, due to the differential downstream pathways, part of which is overlapping and part of which is unique, they exert different effects. It should be noted that the peculiarity of the behavioural effects are also context-dependent in that certain effects depend on the naïve (IN stimulation) versus trained status (FN inhibition) of the animal (e.g., when looking at the impact of stimulation versus that of inhibition). This perspective is discussed in the section 'Multimodular control of sensorimotor tasks and functional implications', line 547-578 in the manuscript.

Reviewer #2:

My previous review when this paper was under consideration for [REDACTED] highlighted a few areas where the manuscript could be improved, namely:

- Doing a causal manipulation (electrical or opto stimulation) in FN to determine whether activation of these neurons can directly drive an eyeblink.*
- Doing a chronic lesion in FN to see if the deficits in CRs and URs remain to rule out that the deficits were due to acute perturbations of the circuit dynamics.*
- Toning down the language about the strength of the trial-by-trial correlations, given that the number of neurons with significant correlations is low.*
- Addressing the (unexpected) result that inhibiting FN caused modulation of activity in IN.*
- Discussing more fully the high level idea the authors are proposing that the FN "dynamically modulates muscle tone" during DEC.*

While it would have been easier to evaluate how well the authors addressed these points (and the more minor ones) if they had tracked changes in the manuscript document or pointed to specific pages/line numbers in their rebuttal (I acknowledge that this may have been at least partially due to the transfer of the submission from Nat Neuro to Nat Comm), I am satisfied that the authors addressed them well enough to warrant publication in Nature Communications.

We thank the reviewer again for these constructive comments as well as the support for publication in Nature Communications. In the new version of the manuscript, we have now also added the chronic lesion experiments and revised the manuscript in track changes.

However, there are a few remaining points I think the authors should address before publication to improve the readability and completeness of the manuscript.

1. Figure 1: I may be missing some critical detail but I'm having trouble understanding why a portion of "facilitated" neurons is negatively correlated with CR amplitude (Fig 1h). Isn't the authors' definition of a "facilitated" neuron that it increases activity during the CR? Is it really the case that some neurons fire a bunch for really small CRs and then fire less strongly (but still above baseline) for bigger CRs? This would be pretty strange but interesting if true. I found this part very confusing. Please clarify.

It is indeed the case. We have recorded from 5 neurons that increased their firing rates during CR, but had a negative correlation with the CR amplitudes. The example neuron in Fig. 1h illustrates larger increases in firing rate when this mouse had smaller CR amplitudes (fitting $y = -0.27x + 77.06$). Below we show 3 raw traces

of the same neuron to further illustrate our results. From top to bottom trials, the mouse had small, medium and big CRs, and strong, medium and weak spike facilitation, respectively.

2. Results related to Fig 3, page 8, second paragraph: Are the n's reported (8 and 1) only those neurons that were significantly correlated? Out of how many total tested? Please clarify.

We have recorded 23 PCs with simple spike suppression (page 8 line 188-190), and among these cells 8 had significant trial-by-trial correlation (page 8 line 203-205). We have recorded 26 PCs with simple spike facilitation (page 8 line 191-193), and only 1 cell had trial-by-trial correlation (page 8 line 207-210). These numbers are now highlighted in the manuscript (page 8) and figure legends (page 41-42 for Fig. 3; supplementary information page 11-12 for Fig. S7).

3. How many of the suppressed PCs had short latency CpxUS? Did any of the facilitated PCs have short latency CpxUS? The authors mostly focus on the CS-related complex spike but the US-related complex spike is the canonical IO signal that has been used for defining eye blink-related cortex zones so I would like to see a little more treatment of this signal in the Results.

Among the suppressed PCs, 10 out of 23 cells had short-latency CpxUS, and for the facilitated PCs, this fraction was 15/26. We have now clarified these fractions in the manuscript (page 9, line 224-226) and supplementary Fig. 8b.

4. While I agree with the authors' point about the limitations of chronic lesions, I could just as easily apply a similar logic to disruptions of circuit dynamics during acute manipulations, so I don't find their rebuttal of this point totally compelling. Is there any other data they can provide to make the case that the particular part of the FN they have studied is uniquely contributing to DEC, rather than the alternative that "disrupting a major cerebellar output causes general circuit dysfunction"? I'm not proposing that they do additional experiments but just hoping that they have some existing data that could address the point. Perhaps from "misses", where they manipulated a different part of the cerebellum and didn't abolish CRs/URs? In my opinion, lack of these data should not prevent this manuscript, which was otherwise a very technically rigorous set of experiments, from being published, but I think it would at least warrant the authors addressing this caveat in the Discussion. And if they have the data it would greatly strengthen their conclusions.

In line with the reviewer's request, we have now chronically lesioned the FN (ipsilateral to the trained eye) in well-trained mice and tested their post-lesion CR performance for 3 days (new Fig. 5j-o). Mice with FN lesions exhibited significant lower CR-trial probability and smaller CR amplitude, compared with either the sham operation group or to their own pre-lesion performance. These deficits lasted for at least 3 days without clear recovery. Therefore, our new chronic lesion data confirm the involvement of FN circuits in DEC, and further strengthen our conclusions.

Reviewer #3:

In the previous round of reviews, all three reviewers shared similar concerns about the functional relevance of the FN pathway for eyeblink conditioning. These concerns included:

- an extensive literature exists which argues that the IN pathway is sufficient and the FN is not needed for CR's. To address this, the authors have "analyzed the activity of FN neurons and vermal PCs in CR trials and non-CR-trials, as well as during spontaneous blinks (Supplementary Fig. 2, 5)." And argue that these results, "suggest that CS and US inputs to the vermis-FN module are not engaged during spontaneous blinks," (OK), "but are crucial for controlling eyelid closure during the task...". I don't see how these data speak to the FN being "crucial".

Our data (Supplementary Fig. 10) are in line with previous literature in that the IN pathway is the key motor driver for eyeblink in DEC. However, we have presented a compelling case, through various recordings, acute and chronic manipulations, and circuit mapping, that FN module is, next to the IN, also needed for DEC. Thus our data implies that the original conclusion that the IN modulation is sufficient might be incorrect. We think that the issue of sufficiency is indeed a very difficult topic in general. For example, one might conclude from the studies by Heiney et al., 2014 (JN) and our own by ten Brinke et al., 2017 (eLife) that the ability to manipulate the CRs by optogenetic stimulation and inhibition of the cerebellar cortex that connects with the IN indicates that the IN circuitry is sufficient to drive DEC. However, in both studies mentioned above, the FN pathways might in principle have been intact in these experiments. Since the authors at the time (and this includes ourselves) assumed that the FN did not have any role, we were excluding the possibility of parallel roles of modules in the IN and FN circuitry and we all felt that the optogenetic manipulation of the IN modular circuitry showed its sufficiency, while we were forgetting about the potential roles of other circuitries that were left intact just as the machinery of the eyelids themselves was left intact. In other words, one can also not state that the IN circuitry is sufficient, because after all you also need the brainstem neurons and motoneurons that innervate the eyelids as well as the eyelids themselves. Together with our ignorance of the potential role of the FN circuitry, we feel it is incorrect to conclude that the IN circuitry was sufficient for driving a complete sensorimotor behaviour. For clarity, this is a completely different question as to whether the IN is essential or whether the FN is sufficient. As stated above, we completely agree with this Reviewer that IN is essential and in fact we also agree that the IN is the motor command driver. Moreover, we also conclude that the FN is not sufficient (albeit necessary as now highlighted with the proper control experiments).

The reviewer also clearly agrees that our study is in line with the previous observations from Wang, Medina and colleagues (from previous comments), as well as Hesslow and colleagues, showing that eyeblink related signals can in principle be

found outside the simplex lobule, and thereby IN circuitry. In that regard we are unsure whether the reviewer views our result as expected or surprising.

By analysing the activity of FN neurons and vermal PCs during CR trials, non-CR trials and spontaneous blinks, we have indeed reached the conclusion that a population of vermis-FN neurons show task-specific modulation for DEC (see page 5, line 112-113). In our original response to reviewer 2 we stated ‘these results suggest that CS and US inputs to the vermis-FN module are not engaged during spontaneous blinks, but are crucial for controlling eyelid closure during the task.’ Subsequently, we further substantiated its crucial role with various recordings and manipulations as presented in the rest of the study. Following the request of the reviewers, we have now also performed the chronic FN lesion experiments and the results again completely support the crucial role of FN in DEC (see below). So taken all together, we hope that our new experiments as well as our clarification convince the reviewer about the importance of the FN module in associative eyeblink conditioning (even though the FN is indeed also, just like the IN module, not sufficient). Finally, we would like to say upfront to this reviewer that we ourselves, as fervent advocates of the role of the IN module, were also puzzled by our initial observations. Hence, this explains why we did many more mechanistic and control experiments to confirm our findings on the FN module, basically repeating many of the experiments that were done on the IN module by many groups for decades.

- the possibility that the vermal/FN signals are not specifically related to eyelid movements (Reviewers 1 and 3). Here the authors make some arguments about response latencies but ultimately admit that their data “at least raise the possibility” that these signals are related to eyelid movements.

This point has been addressed in our previous rebuttal to Reviewer 1, but we would like to clarify it better. As we recorded vermal PCs/FN neurons with multi-channel probes, it is fairly certain that we included eyeblink related neurons as well as neurons that were unrelated to the task. Therefore we carried out careful analysis on the trial-by-trial correlation, the comparison of CR/non-CR trials, the temporal relationships between activity and behaviour, and the identification of short latency complex spikes to clarify which neurons are most likely related to the task. These results suggest that the activity of FN neurons is, at least partially, associated with eyelid movements during DEC (page 5, line 112-113). However, we do not exclude the possibility that some FN neurons also play other roles during DEC. Interestingly, here too, we feel that in the past, both others and we ourselves might have overestimated the specific correlations between neuronal activity in the IN module and DEC. Evidence is emerging that these correlations also reflect other parts of the face or body that might be engaged in the same defensive-like response.

- the use of only acute inhibition (muscimol, DREADDs) that could acutely disrupt

activity in cerebellar target regions, and no chronic manipulations (Reviewer 2). The authors expressed concern that if they did these experiments, the results might be difficult to interpret. But given the valid concerns about the acute manipulations, we can only really judge that once the experiments are done and the data presented.

We agree with Reviewers 2 and 3 on this point. Following the suggestion of these reviewers, we have now lesioned the FN (ipsilateral to the trained eye) in well-trained mice and tested their post-lesion CR performance for 3 days (new Fig. 5j-o). Mice with an FN lesion exhibited significant lower CR-trial probability and smaller CR amplitude, compared with either the sham operation group or to their own pre-lesion performance. These deficits last for at least 3 days without clear recovery. Therefore, our new chronic lesion data confirm the involvement of FN circuits in DEC, and further strengthen our conclusion.

- the weak correlation between the kinematics of the eyelid closure and the neural signals (Reviewers 2 and 3). The authors “toned down the claim by using ‘significant correlation’ instead of ‘strong/clear correlation’,” and they argue in the rebuttal that the small percentages of modulated neurons are justified by an entirely unrelated recent study. They have not really responded to my concern about a lack of analysis or demonstration of trial-to-trial correlations of the timing of the responses and the behavior.

We have argued in the previous rebuttal that 1) the proportion of correlated cells is not small considering our unbiased multi-channel recording approach. In fact, we showed that despite this unbiased approach 17% of FN neurons and 35% of the related PCs had trial-by-trial correlations. Thus, as there are many more functions controlled by FN, we respectfully disagree with the view that these correlations are weak. 2) We cited the Steinmez *et al* study to stress the point that the fraction of task-related neurons are inherently low in high density multi-channel recordings. Our conclusion is also in line with a recent study revealing that FN comprises 5 heterogeneous groups, which could contribute to an even large number of different functions (Fujita et al., 2020 eLife).

Finally, analysing the trial-to-trial correlations of the timing aspects is methodologically challenging. The trial-to-trial variability of CR onset is typically 2030 ms, which is within the same range as the single inter-spike interval of FN neurons and PCs. Accurately determining the single trial spike onset timing of single neurons is therefore questionable at this timescale. We therefore performed our timing analysis at the population level (Fig. 1 j-l, Fig. 3 g-h).

- the nature of the FN modulation of learning (all Reviewers). The authors object to the characterization of the FN role as ‘permissive,’ even though it was their exact

language in the original manuscript. They have changed the language, but in my view these concerns, raised as central points by all three reviewers, remain.

As stated in the previous rebuttal, our revision experiments presented evidence that vermal PCs and FN neurons are not merely permissive. Chemogenetic and optogenetic suppression of FN neurons caused persistent learning deficits that were still present after we omitted the inhibition. Consistently, chronic lesions of FN also resulted in long-lasting deficits of CR performance. Based on these data we think that the FN pathway is actively engaged in online modulation of CR performance rather than merely being a passive permissive hub. We have revised the wording of our manuscript accordingly.

REVIEWERS' COMMENTS

Reviewer #1 (Remarks to the Author):

I am satisfied with the revisions to the manuscript and recommend publication.

Reviewer #2 (Remarks to the Author):

I'm still a bit confused by this story and am not quite sure how to think about it, which is apparently a sentiment shared by the other reviewers as well. But I don't think that's necessarily a bad thing. The authors have clearly made an effort to address many of the major comments and concerns, and the manuscript is much improved as a result. I think the study is scientifically sound--putting aside some methodological oddities/caveats like the strong modulation of IN during optogenetic inhibition of FN. I believe that the remaining issues with the study are many of the same ones that the field is currently grappling with, such as whether neural activity that is correlated with behavior can be taken as evidence that the activity controls the behavior and whether acutely or chronically disrupting a circuit can tell you what the circuit normally does. A single study will not be able to resolve the deeper questions about the roles of different cerebellar modules in behavior, but I think this study adds important information to the debate. I certainly find it interesting.

However, before publication I think the authors could still do a better job making clear in the text the total number of neurons that were tested for each reported result and statistical test so the reader can more easily understand the proportions. In reference to my comment in the previous review, if they tested 23 PCs and 8 showed significant positive correlations, it would be much easier for the reader to judge the importance of the result if they report the proportion $n=8/23$ instead of just $n=8$. Otherwise, it's not clear if they tested the entire dataset of 23 PCs that were mentioned in the previous paragraph or had to, for instance, omit some because they didn't have enough trials to analyze the trial-by-trial correlations. The same comment also applies to the FN neurons. When you start with a dataset of 162 neurons and end up with a handful in each category it's helpful to more explicitly spell out how you arrived at the eventual numbers.

Rebuttal letter

Reviewer #2:

I'm still a bit confused by this story and am not quite sure how to think about it, which is apparently a sentiment shared by the other reviewers as well. But I don't think that's necessarily a bad thing. The authors have clearly made an effort to address many of the major comments and concerns, and the manuscript is much improved as a result. I think the study is scientifically sound--putting aside some methodological oddities/caveats like the strong modulation of IN during optogenetic inhibition of FN. I believe that the remaining issues with the study are many of the same ones that the field is currently grappling with, such as whether neural activity that is correlated with behavior can be taken as evidence that the activity controls the behavior and whether acutely or chronically disrupting a circuit can tell you what the circuit normally does. A single study will not be able to resolve the deeper questions about the roles of different cerebellar modules in behavior, but I think this study adds important information to the debate. I certainly find it interesting.

However, before publication I think the authors could still do a better job making clear in the text the total number of neurons that were tested for each reported result and statistical test so the reader can more easily understand the proportions. In reference to my comment in the previous review, if they tested 23 PCs and 8 showed significant positive correlations, it would be much easier for the reader to judge the importance of the result if they report the proportion $n=8/23$ instead of just $n=8$. Otherwise, it's not clear if they tested the entire dataset of 23 PCs that were mentioned in the previous paragraph or had to, for instance, omit some because they didn't have enough trials to analyze the trial-by-trial correlations. The same comment also applies to the FN neurons. When you start with a dataset of 162 neurons and end up with a handful in each category it's helpful to more explicitly spell out how you arrived at the eventual numbers.

We have followed the advice from the reviewer and further clarified the number of cells in the manuscript. See page 5, 6, 8, 9 and legends of figure 1, 3. We agree that our findings contribute to the current debate about the roles cerebellar modules in controlling sensorimotor behaviour and sincerely hope that our work will pave the way for future studies.